# ONLINE TARGET Q-LEARNING WITH REVERSE EXPERIENCE REPLAY: EFFICIENTLY FINDING THE OPTIMAL POLICY FOR LINEAR MDPS

**Naman Agarwal, Prateek Jain, Dheeraj Nagaraj, Praneeth Netrapalli**
Google Research
`{namanagarwal,prajain,dheeraj,pnetrapalli}@google.com`

**Syomantak Chaudhuri**
University of California, Berkeley
`syomantak@berkeley.edu`

## ABSTRACT

Q-learning is a popular Reinforcement Learning (RL) algorithm which is widely deployed with function approximation (Mnih et al., 2015). In contrast, existing theoretical results are pessimistic about Q-learning. For example, Q-learning does not converge even with linear function approximation for linear MDPs ((Baird, 1995)) and even for tabular MDPs with synchronous updates, Q-learning has sub-optimal sample complexity (Li et al., 2021; Azar et al., 2013). The goal of this work is to bridge the gap between practical success of Q-learning and the relatively pessimistic theoretical results. The starting point of our work is the observation that in practice, Q-learning is used with two important modifications: (i) training with two networks, called online network and target network simultaneously (online target learning, or OTL) , and (ii) experience replay (ER) (Mnih et al., 2015). While they play a significant role in the practical success of Q-learning, a thorough theoretical understanding of how these two modifications improve the convergence behavior of Q-learning has been missing in literature. By carefully combining Q-learning with OTL and *reverse* experience replay (RER) (a form of experience replay), we present novel methods Q-Rex and Q-RexDaRe (Q-Rex+ data reuse). We show that Q-Rex efficiently finds the optimal policy for linear MDPs (or more generally for MDPs with zero inherent Bellman error with linear approximation (ZIBEL)) and provide non-asymptotic bounds on sample complexity – the first such result for a Q-learning method for this class of MDPs under standard assumptions. Furthermore, we demonstrate that Q-RexDaRe in fact achieves near optimal sample complexity in the tabular setting, improving upon the existing results for vanilla Q-learning.

## 1 INTRODUCTION

Reinforcement Learning (RL) has been shown to be highly successful in practice for a variety of long term decision making problems (Mnih et al., 2015). Several classical works have studied RL methods like TD-learning, Q-learning and their variants (Sutton & Barto, 2018; Bertsekas, 2011; Borkar & Meyn, 2000; Sutton, 1988; Tsitsiklis & Van Roy, 1997; Watkins & Dayan, 1992; Watkins, 1989) but the guarantees are mostly asymptotic and therefore do not sufficiently answer important questions that are relevant to practitioners who struggle with constraints on the number of data points and the computation power. Recent works provide non-asymptotic results for a variety of important settings (Kearns & Singh, 1999; Beck & Srikant, 2012; Qu & Wierman, 2020; Ghavamzadeh et al., 2011; Bhandari et al., 2018; Chen et al., 2020; 2019; Dalal et al., 2018a;b; Doan et al., 2020; Gupta et al., 2019; Srikant & Ying, 2019; Weng et al., 2020; Yang & Wang, 2019; Zou et al., 2019a).

Despite a large body of work, several aspects of fundamental methods like Q-learning (Watkins & Dayan, 1992) are still ill-understood. Q-learning's simplicity and the ability to learn from off-policy

data makes it widely applicable. However, theoretical analyses show that even with *linear function approximation* and when the approximation is *exact*, Q-learning can fail to converge even in simple examples (Baird, 1995; Boyan & Moore, 1995; Tsitsiklis & Van Roy, 1996). Furthermore, even in the simple case of tabular RL with synchronous updates, Q-learning is known to have sub-optimal sample complexity (Wainwright, 2019a; Li et al., 2021).

However, Q-learning has seen tremendous practical success when deployed with "heuristic" modifications like experience replay (ER) and online target learning (OTL). ER is used to alleviate the issues arising due highly dependent samples in an episode whereas OTL helps stabilize the Q iteration. Mnih et al. (2015) conducted extensive experiments to show that *both* these techniques, are essential for the success of Q-learning. But, existing analyses for ER with Q-learning either require stringent assumptions (Carvalho et al., 2020) to ensure convergence to a good Q value, or assume that ER provides i.i.d. samples which might not hold in practice (Fan et al., 2020; Carvalho et al., 2020). In this paper, we attempt to bridge the gap between theory and practice, by rigorously investigating how Q-learning performs with these practical heuristics. We thus introduce two model free algorithms: Q-Rex and Q-RexDaRe that combine the standard Q-learning with OTL and *reverse* experience replay (RER). RER is a form of ER which was recently studied to unravel spurious correlations present while learning form Markovian data in the context of system identification (Jain et al., 2021b). We show that OTL stabilizes the Q value by essentially serving as a variance reduction technique and RER unravels the spurious correlations present in the Markovian data to remove inherent biases introduced in vanilla Q learning.

These simple modifications have surprisingly far-reaching consequences. Firstly, this allows us to show that unlike vanilla Q-learning, Q-Rex finds the optimal policy for MDPs with an exact linear function representation of the Bellman operator. and allows us to derive non-asymptotic sample complexity bounds. In the *tabular setting*, Q-Rex even with asynchronous data is able to match the best known bounds for Q-learning with synchronous data. Its variant Q-RexDaRe , which reuses old samples, admits nearly optimal sample complexity for recovering the optimal Q-function in the tabular setting. Previously, only Q-learning methods with explicit variance-reduction techniques (not popular in practice) (Wainwright, 2019b; Li et al., 2020b) or model based methods (Agarwal et al., 2020; Li et al., 2020a) were known to achieve such a sample complexity bound. Our experiments show that when the algorithmic parameters are chosen carefully, Q-Rex and its variants outperform both vanilla Q-learning and OTL+ER+Q-learning with the same parameters (see Appendix A).

To summarize, in this work, we study Q-learning with practical heuristics like ER and OTL, and propose two concrete methods Q-Rex and Q-RexDaRe based on OTL and reverse experience replay – a modification of the standard ER used in practice. We show that Q-Rex is able to find the optimal policy for ZIBEL MDPs, with a strong sample complexity bound which is the first such result for Q-learning. We also show that Q-RexDaRe obtains nearly optimal sample complexity for the simpler tabular setting despite not using any explicit variance reduction technique. See Table 1 for a comparison of our guarantees against the state-of-the-results for the tabular setting.

**Organization** We review related works in next subsection. In Section 2 we develop the MDP problem which we seek to solve and present our algorithm, Q-Rex in Section 3. The main theoretical results are presented in Section 4. We present a brief overview of the analysis in Section 5 and present our experiments in Section A. We provide minimax lower bounds for the asynchronous tabular setting in Section K. Most of the formal proofs are relegated to the appendix.

## 1.1 RELATED WORKS

**Tabular Q-learning** Tabular MDPs are the most basic examples of MDPs where the state space ($\mathcal{S}$) and the action space ($\mathcal{A}$) are both finite and the Q-values are represented by assigning a unique co-ordinate to each state-action pair. This setting has been well studied over the last few decades and convergence guarantees have been derived in both asymptotic and non-asymptotic regimes for popular model-free and model-based algorithms. Azar et al. (2013) shows that the minimax lower bounds on the sample complexity of obtaining the optimal Q-function up-to $\epsilon$ error is $\frac{|\mathcal{S}||\mathcal{A}|}{(1-\gamma)^3\epsilon^2}$, where $\gamma$ is the discount factor. Near sample-optimal estimation is achieved by several model-based algorithms (Agarwal et al., 2020; Li et al., 2020a) and model-free algorithms like variance reduced Q-learning (Wainwright, 2019b; Li et al., 2020b). (Li et al., 2021) also shows that vanilla Q-learning with standard step sizes, even in the synchronous data setting – where transitions corresponding to each state

| Paper | Algorithm | Data Type | Sample Complexity |
|---|---|---|---|
| (GHAVAMZADEH ET AL., 2011) | SPEEDY Q-LEARNING | SYNCHRONOUS | $\frac{|\mathcal{S}||\mathcal{A}|}{\epsilon^2(1-\gamma)^4}$ |
| (WAINWRIGHT, 2019B) | VARIANCE REDUCED Q-LEARNING | SYNCHRONOUS | $\frac{|\mathcal{S}||\mathcal{A}|}{\epsilon^2(1-\gamma)^3}$ |
| (LI ET AL., 2020B) | VARIANCE REDUCED Q-LEARNING | ASYNCHRONOUS | $\frac{1}{\mu_{\min}\epsilon^2(1-\gamma)^3}$ |
| (LI ET AL., 2020B) | Q-LEARNING | ASYNCHRONOUS | $\frac{1}{\mu_{\min}\epsilon^2(1-\gamma)^5}$ |
| (LI ET AL., 2021) | Q-LEARNING | SYNCHRONOUS | $\frac{|\mathcal{S}||\mathcal{A}|}{\epsilon^2(1-\gamma)^4}$ |
| THIS WORK, THEOREM 2 | Q-LEARNING+ OTL + RER (Q-REX) | ASYNCHRONOUS | $\frac{|\mathcal{S}||\mathcal{A}|}{\epsilon^2(1-\gamma)^4}$ |
| THIS WORK, THEOREM 3 | Q-REX+ DATA-REUSE (Q-REXDARE) | ASYNCHRONOUS | $\frac{\max(\bar{d},\frac{1}{\epsilon^2})}{\mu_{\min}(1-\gamma)^3}$ |

Table 1: Comparison of tabular Q-learning based algorithms. $\bar{d} \leq |\mathcal{S}|$ is maximum size of support of $P(\cdot|s,a)$. In the case of asynchronous setting, $\frac{1}{\mu_{\min}}$ is roughly equivalent to $|\mathcal{S}||\mathcal{A}|$ in the synchronous setting. We use the color green to represent results with optimal dependence on $(1-\gamma)^{-1}$.

action pair are sampled independently at each step – suffers from a sample complexity of $\frac{|\mathcal{S}||\mathcal{A}|}{(1-\gamma)^4\epsilon^2}$ and the best known bounds in the asynchronous setting – where data is derived from a Markovian trajectory and only one Q value is updated in each step – is $\frac{|\mathcal{S}||\mathcal{A}|}{(1-\gamma)^5\epsilon^2}$. These results seem unsatisfactory since $\gamma \in (0.99, 1)$ in most practical applications. In contrast, our algorithm Q-Rex with *asynchronous* data has a sample complexity that matches Q-learning bound with *synchronous* data and its data-efficient variant Q-RexDaRe has near minimax optimal sample complexity (see Table 1). For details on model based algorithms, and previous works with sub-optimal guarantees we refer to (Agarwal et al., 2020; Li et al., 2020b). We note that the lower bounds apply only to the synchronous case (i.e, when every state-action pair is sampled at every step). We provide minimax lower-bounds which show that the bound is tight in the asynchronous case too (see Theorem 5 in Section K), where $|\mathcal{S}||\mathcal{A}|$ in the synchronous case of (Azar et al., 2013) is replaced by $\frac{1}{\mu_{\min}}$.

**Q-learning with Linear Function Approximation** Since tabular Q-learning is intractable in most practical RL problems due to a large state space $\mathcal{S}$, function approximation is deployed. Linear function approximation is the simplest such case where the Q-function is approximated with a linear function of the 'feature embedding' associated with each state-action pair. However, Q-learning can be shown to diverge even in the simplest cases as was first noticed in (Baird, 1995), which also introduced residual gradient methods which converged rather slowly but provably. We will only discuss recent works closest to our work and refer the reader to (Carvalho et al., 2020; Yang & Wang, 2019) for a full survey of various works in this direction. SARSA is the on-policy control variant of Q-learning where the challenge is to explore the state-space while learning the optimal policy. Unlike Q-learning, SARSA is inherently stable due to its on-policy nature (Gordon, 2000). Therefore, we do not compare our results to the results of on-policy control algorithms like SARSA. We refer to (Zou et al., 2019b; Perkins & Precup, 2002; Melo et al., 2008) for further details.

Yang & Wang (2019) consider MDPs with approximate linear function representation. They require additional assumptions like finite state-action space and existence of *known* anchor subsets which might not hold in practice. Our results on the other hand hold with standard assumptions, with *asynchronous* updates and can handle *infinite* state-action spaces (see Theorem 1). Similarly, Chen et al. (2019) consider Q-learning with linear function approximation which need not be exact. But the result requires the restrictive assumption that the offline policy is close to the optimal policy. In contrast, we consider the less general but well-studied case of MDPs with zero inherent Bellman error and provide global convergence without restrictive assumptions on the behaviour policy.

Under the most general conditions Maei et al. (2010) present the Greedy-GQ algorithm which converges to a point asymptotically instead of diverging. Similar results are obtained by Carvalho et al. (2020) for Coupled Q-learning, a 2-timescale variant of Q-learning which uses a version of OTL and ER[1]. This algorithm experimentally resolves the popular counter-examples provided by (Tsitsiklis & Van Roy, 1996; Baird, 1995). However, the value function guarantees in Carvalho et al. (2020,

---

[1]The version of ER used in Carvalho et al. (2020) makes the setting completely *synchronous* as opposed to the *asynchronous* setting considered by us.

Theorem 2) (albeit without sample complexity guarantees) requires very stringent assumptions and even in the case of tabular Q-learning might not converge to the optimal policy.

**Experience Replay and Reverse Experience Replay**   Reinforcement learning involves learning on-the-go with highly correlated correlated data. Iterative learning algorithms like Q-learning can sometimes get coupled to the Markov chain resulting in sub-optimal convergence. Experience replay (ER) was introduced in order to mitigate this drawback (Lin, 1992) – here a large FIFO buffer of a fixed size stores the streaming data and the learning algorithm samples a data point uniformly at random from this buffer at each step. This makes the samples look roughly i.i.d., thus breaking the harmful correlations. Reverse experience replay (RER) is a form of ER which stores data in a buffer but processes the data points in the reverse order as stored in the buffer. This was introduced in entirely different contexts by (Rotinov, 2019; Jain et al., 2021b;a). In this work, we note that reverse order traversal endows a super-martingale structure which yields the strong concentration result in Theorem 4, which is not possible with forward order traversal (see (Jain et al., 2021b, Section 3.1) for a brief demonstration of this fact). We can also look at RER is through the lens of Dynamic programming (Bertsekas, 2011) – where the value function is evaluated backwards starting from time $T$ to time 1 - similar to how RER bootstraps present to the future.

In the context of reinforcement learning, works like Bhandari et al. (2018); Zou et al. (2019b) obtain finite time convergence guarantees for RL algorithms under the mixing assumptions just like this work. The strategy followed in these works is that if two samples are $\tilde{O}(\tau_{\text{mix}})$ time apart, then they are approximately independent and thus analysis for i.i.d. data can be applied. The sample complexity to obtain $\epsilon$ error here is $O(\frac{\tau_{\text{mix}}}{\epsilon^2})$. Note that this is no better than keeping one every $\tau_{\text{mix}}$ samples and throwing away the rest and under general mis-specified linear function representation, we might not be able to do any better (Bresler et al., 2020). In this work, we show that when the linear approximation is well specifed (ZIBEL), we can use RER to obtain a sample complexity of $O(\tau_{\text{mix}} + \frac{1}{\epsilon^2})$, where the mixing time serves as a cut-off.

**Online Target Learning**   OTL (Mnih et al., 2015) maintains two different Q-values (called online Q-value and target Q-value) where the target Q-value is held constant for some time and only the online Q-value is updated by 'bootstrapping' to the target. After a number of such iterations, the target Q-value is set to the current online Q value. OTL thus attempts to mitigate the destabilizing effects of bootstrapping by removing the 'moving target'. This technique has been noted to allow for an unbiased estimation of the bellman operator (Fan et al., 2020) and when trained with large batch sizes is similar to the well known neural fitted Q-iteration (Riedmiller, 2005).

## 2   PROBLEM SETTING

**Markov Decision Process**   We consider infinite horizon, time homogenous Markov Decision Processes (MDPs) and we denote an MDP by MDP($\mathcal{S}, \mathcal{A}, \gamma, P, R$) where $\mathcal{S}$ is the state space, $\mathcal{A}$ is the action space, $\gamma \in [0, 1)$ is the discount factor, $P(s'|s, a)$ is the probability of transition to state $s'$ from the state $s$ on action $a$. We assume that $\mathcal{S}$ and $\mathcal{A}$ are compact subsets of $\mathbb{R}^n$ (for some $n \in \mathbb{N}$). $R : \mathcal{S} \times \mathcal{A} \to [0, 1]$ is the deterministic reward associated with every state-action pair.

We will think of an MDP as an agent that is aware of its current state and it can choose the action to be taken. Suppose the agent takes action $\pi(s)$, where $\pi : \mathcal{S} \to \mathcal{A}$, at state $s \in \mathcal{S}$, then $P$ along with the 'policy' $\pi$ induces a Markov chain over $\mathcal{S}$, whose transition kernel is denoted by $P^\pi$. We write the $\gamma$-discounted value function of the MDP starting at state $s$ to be:

$$V(s, \pi) = \mathbb{E}[\sum_{t=0}^{\infty} \gamma^t R(S_t, A_t)|S_0 = s, A_t = \pi(S_t)\forall t]. \tag{1}$$

It is well-known that under mild assumptions, there exists at least one optimal policy $\pi^*$ such that the value function $V(s, \pi)$ is maximized for every $s$ and that there is an optimal Q-function, $Q^* : \mathcal{S} \times \mathcal{A} \to \mathbb{R}$, such that one can find the optimal policy as $\pi^*(s) = \arg\max_{a \in \mathcal{A}} Q^*(s, a)$, optimal value function as $V^*(s) = \max_{a \in \mathcal{A}} Q^*(s, a)$ and it satisfies the following fixed point equation.

$$Q^*(s, a) = R(s, a) + \gamma \mathbb{E}_{s' \sim P(\cdot|s,a)}[\max_{a' \in \mathcal{A}} Q^*(s', a')] \qquad \forall (s, a) \in \mathcal{S} \times \mathcal{A}. \tag{2}$$

(2) can be alternately viewed as $Q^*$ being the fixed point of the Bellman operator $\mathcal{T}$, where

$$\mathcal{T}(Q)(s, a) = R(s, a) + \gamma \mathbb{E}_{s' \sim P(\cdot|s,a)}[\max_{a'} Q(s', a')].$$

The basic task at hand is to estimate $Q^*(s, a)$ from a single trajectory $(s_t, a_t)_{t=1}^T$ such that $s_{t+1} \sim P(\cdot|s_t, a_t)$ along with rewards $(r_t)_{t=1}^T$, where $r_1, \ldots, r_T$ are random variables such that $\mathbb{E}[r_t|s_t = s, a_t = a] = R(s, a)$. We refer to Section B for rigorous definitions.

**Q-learning**  Since the transition kernel $P$ (and hence the Bellman operator $\mathcal{T}$) is often unknown in practice, Equation (2) cannot be directly used to to estimate the optimal Q-function. To this end, we resort to estimating $Q^*$ using observations from the MDP. An agent traverses the MDP and we obtain the state, action, and the reward obtained at each time step. We assume the *off-policy* setting which means that the agent is not in our control, i.e., it is not possible to choose the agent's actions; rather, we just observe the state, the action, and the corresponding reward. Further, we assume that the agent follows a time homogeneous policy $\pi(s)$ for choosing its action at state $s$.

Given a trajectory $\{s_t, a_t, r_t\}_{t=1}^T$ generated using some unknown behaviour policy $\pi$, we aim to estimate $Q^*$ in a *model-free* manner - i.e, estimate $Q^*$ without directly estimating $P$. We further assume that the trajectory is given to us as a data stream so we can not arbitrarily fetch the data for any time instant. A popular method to estimate $Q^*$ is using the Q-learning algorithm. In this online algorithm, we maintain an estimate of $Q^*(s, a)$ at time $t$, $Q_t(s, a)$ and the estimate is updated at time $t$ for $(s, a) = (s_t, a_t)$ in the trajectory. Formally, with step-sizes given as $\{\eta_t\}$, Q-learning performs the following update at time $t$,

$$Q_{t+1}(s_t, a_t) = (1 - \eta_t)Q_t(s_t, a_t) + \eta_t \left[ r_t + \gamma \max_{a' \in \mathcal{A}} Q_t(s_{t+1}, a') \right]$$

$$Q_{t+1}(s, a) = Q_t(s, a) \quad \forall (s, a) \neq (s_t, a_t). \tag{3}$$

In this work, we focus on two special classes of MDPs which are popular in literature.

**Linear Markov Decision Process**  Linear MDPs (Jin et al., 2020) is a popular example of exact linear approximation for which statistically and computationally tractable algorithms are available. We use the definition from Jin et al. (2020), stated as Definition 1.

**Definition 1.** *An MDP$(\mathcal{S}, \mathcal{A}, \gamma, \mathbb{P}, R)$ is a linear MDP with feature map $\phi : \mathcal{S} \times \mathcal{A} \to \mathbb{R}^d$, if*

1. *there exists a vector $\theta \in \mathbb{R}^d$ such that $R(s, a) = \langle \phi(s, a), \theta \rangle$, and*

2. *there exists $d$ unknown (signed) measures over $\mathcal{S}$ $\beta(\cdot) = \{\beta_1(\cdot), \ldots, \beta_d(\cdot)\}$ such that the transition probability $P(\cdot|s, a) = \langle \phi(s, a), \beta(\cdot) \rangle$.*

In the rest of this paper, in the tabular setting we assume that the dimension $d = |\mathcal{S} \times \mathcal{A}|$ and we use a one hot embedding where we map $(s, a) \to \phi(s, a) = e_{s,a}$, a unique standard basis vector. It is easy to show that this system is a linear MDP (Jin et al., 2020) and Q-learning in this setting reduces to the standard tabular Q-learning (3). However, when the assumption of a tabular MDP allows us to obtain stronger results, we will present the analysis separately.

**Inherent Bellman Error**  There is another widely studied class of MDPs which admit a good linear representation (Zanette et al., 2020; Munos & Szepesvári, 2008; Szepesvári & Smart, 2004).

**Definition 2.** *(ZIBEL MDP) For an $\mathcal{M} = MDP(\mathcal{S}, \mathcal{A}, \gamma, \mathbb{P}, R)$ with a feature map $\phi : \mathcal{S} \times \mathcal{A} \to \mathbb{R}^d$, we define the inherent Bellman error (IBE($\mathcal{M}$)) as:*

$$\sup_{\theta \in \mathbb{R}^d} \inf_{\theta' \in \mathbb{R}^d} \sup_{(s,a) \in \mathcal{S} \times \mathcal{A}} \left| \langle \phi(s, a), \theta' \rangle - R(s, a) - \gamma \mathbb{E}_{s' \sim P(\cdot|s,a)} \sup_{a' \in \mathcal{A}} \langle \theta, \phi(s', a') \rangle \right|$$

*If IBE($\mathcal{M}$) = 0, then call this MDP a ZIBEL (zero inherent Bellman error with linear function approximation) MDP.*

The class of ZIBEL MDPs is strictly more general than the class of linear MDPs (Zanette et al., 2020). Both these classes of MDPs have the property that there exists a vector $w^* \in \mathbb{R}^d$ such that the optimal Q-function, $Q^*(s, a) = \langle \phi(s, a), w^* \rangle$ for every $(s, a) \in \mathcal{S} \times \mathcal{A}$, which can be explicity expressed as a function of $\theta, \beta$ and $Q^*$. More generally, they allow us to lift the Bellman iteration to $\mathbb{R}^d$ *exactly* and update our estimates for $w^*$ values directly (Lemmas 3, 4).

Hence, we can focus on estimating $Q^*$ by estimating $\boldsymbol{w}^*$. To this end, the standard Q-Learning approach to learning the Q function can be extended to the linear case as follows:

$$w_{t+1} = w_t + \eta_t \left[ r_t + \gamma \max_{a' \in \mathcal{A}} \langle \phi(s_{t+1}, a'), w_t \rangle - \langle \phi(s_t, a_t), w_t \rangle \right] \phi(s_t, a_t).$$

The above update can be seen as a gradient descent step on the loss function $f(\boldsymbol{w}_t) = (\langle \phi(s_t, a_t), \boldsymbol{w}_t \rangle - \mathsf{target})^2$ where $\mathsf{target} = r_t + \gamma \max_{a'} \langle \phi(s_{t+1}, a'), \boldsymbol{w}_t \rangle$. This update while heavily used in practice, has been known to be unstable and does not converge to $w^*$ in general. The reason often cited for this phenomenon is the presence of the 'deadly triad' of bootstrapping, function approximation, and off-policy learning.

### 2.1 ASSUMPTIONS

We make the following assumptions on the MDPs considered through the paper in order to present our theoretical results.

**Assumption 1.** *The MDP $\mathcal{M}$ has $\mathsf{IBE}(\mathcal{M}) = 0$ (Definition 2), $\|\phi(s, a)\|_2 \leq 1$. Furthermore, $R(s, a) \in [0, 1]$.*

**Assumption 2.** *Let $\Phi := \{\phi(s, a) : (s, a) \in \mathcal{S} \times \mathcal{A}\}$. $\Phi$ is compact, $\mathsf{span}(\Phi) = \mathbb{R}^d$ and $(s, a) \to \phi(s, a)$ is measurable.*

Even when $\mathsf{span}(\Phi) \neq \mathbb{R}^d$, our results hold after we discard the space orthogonal to the span of embedding vectors in Assumption 4 and note that Q-Rex does not update the iterates along $\mathsf{span}(\Phi)^\perp$.

**Definition 3.** *For $r > 0$, let $\mathcal{N}(\Phi, \| \cdot \|_2, r)$ be the $r$-covering number under the standard Euclidean norm over $\mathbb{R}^d$. Define:*

$$C_\Phi := \int_0^\infty \sqrt{\log \mathcal{N}(\Phi, \|\|_2, r)} dr$$

Observe that since $\Phi$ is a subset of the unit Euclidean ball in $\mathbb{R}^d$, $C_\Phi \leq C\sqrt{d}$. However, in the case of tabular MDPs it is easy to show that $C_\Phi \leq C\sqrt{\log d}$.

**Definition 4.** *We define the norm $\| \cdot \|_\phi$ over $\mathbb{R}^d$ by $\|x\|_\phi = \sup_{(s,a)} |\langle \phi(s, a), x \rangle|$.*

This is the natural norm of interest for the problem (Lemmas 2 and 4). We assume the existence of a fixed (random) behaviour policy $\pi : \mathcal{S} \to \Delta(\mathcal{A})$ which selects a random action corresponding to each state. At each step, given $(s_t, a_t) = (s, a)$, $s_{t+1} \sim P(\cdot|s, a)$ and $a_{t+1} \sim \pi(s_{t+1})$. This gives us a Markov kernel over $\mathcal{S} \times \mathcal{A}$ which specifies the law of $(s_{t+1}, a_{t+1})$ conditioned on $(s_t, a_t)$. We will denote this kernel by $P^\pi$. This setting is commonly known as the off-policy *asynchronous* setting. We make the following assumption which is standard in this line of work.

**Assumption 3.** *There exists a unique stationary distribution $\mu$ for the kernel $P^\pi$. Moreover, this Markov chain is exponentially ergodic in the total variation distance with mixing time $\tau_{\mathsf{mix}}$. That is, there exist a constant $C_{\mathsf{mix}}$ for every $t \in \mathbb{N}$*

$$\sup_{x \in \mathcal{S} \times \mathcal{A}} \mathsf{TV}((P^\pi)^t(x, \cdot), \mu) \leq C_{\mathsf{mix}} \exp(-t/\tau_{\mathsf{mix}})$$

*In the tabular setting, we will use the standard definition of $\tau_{\mathsf{mix}}$ instead:*

$$\tau_{\mathsf{mix}} = \inf\{t : \sup_{x \in \mathcal{S} \times \mathcal{A}} \mathsf{TV}((P^\pi)^t(x, \cdot), \mu) \leq 1/4\}.$$

*Here $\mathsf{TV}$ refers to the total variation distance.*

**Assumption 4.** *There exists $\kappa > 0$ such that: $\mathbb{E}_{(s,a) \sim \mu} \phi(s, a) \phi^\top(s, a) \succeq \frac{I}{\kappa}$.*

**Remark 1.** *(Bresler et al., 2020, Theorem 1) shows that even linear regression with Markovian data, zero noise and $\ell^2$ recovery is hard when the condition number $\kappa$ or the mixing time $\tau_{\mathsf{mix}}$ are too large. Hence, our setup of noisy reinforcement learning with $\ell^\infty$ error also requires these quantities to be small. Therefore, Assumptions 3 and 4 are necessary in order to obtain non-trivial bounds.*

In the tabular setting, Assumption 4 manifests itself as $\frac{1}{\kappa} = \mu_{\min} := \min_{(s,a)} \mu(s, a)$ which is also standard (Li et al., 2020b). Whenever we discuss high probability bounds (i.e, probability at least $1 - \delta$), we assume that $\delta \in (0, 1/2)$. Similarly, we will assume that the discount factor $\gamma \in (1/2, 1)$ so that we can absorb $\mathsf{poly}(1/\gamma)$ factors into constants.

---

**Algorithm 1** Q-Rex

---

1: **Input:** learning rates $\eta$, horizon $T$, discount factor $\gamma$, trajectory $X_t = \{s_t, a_t, r_t\}$, Buffer size $B$, Buffer gap $u$, Number of inner loop buffers $N$
2: Total buffer size: $S \leftarrow B + u$, Outer-loop length: $K \leftarrow \frac{T}{NS}$, Initialization $w_1^{1,1} = 0$
3: **for** $k = 1, \ldots, K$ **do**
4:     **for** $j = 1, \ldots, N$ **do**
5:         Form buffer $\mathsf{Buf} = \{X_1^{k,j}, \ldots, X_S^{k,j}\}$, where, $X_i^{k,j} \leftarrow X_{NS(k-1)+S(j-1)+i}$
6:         Define for all $i \in [1, S]$, $\phi_i^{k,j} \triangleq \phi(s_i^{k,j}, a_i^{k,j})$.
7:         **for** $i = 1, \ldots, B$ **do**
8: $$w_{i+1}^{k,j} = w_i^{k,j} + \eta \left[ r_{B+1-i}^{k,j} + \gamma \max_{a' \in \mathcal{A}} \langle \phi(s_{B+2-i}^{k,j}, a'), w_1^{k,1} \rangle - \langle \phi_{B+1-i}^{k,j}, w_i^{k,j} \rangle \right] \phi_{B+1-i}^{k,j}$$
9:         **Option I**: $w_1^{k,j+1} = w_{B+1}^{k,j}$
10:         **Option II**: $w_1^{k,j+1} = \frac{1}{B} \sum_{i=1}^{B} w_{i+1}^{k,j}$
11:     **Option I**: $w_1^{k+1,1} = w_1^{k,N+1}$
12:     **Option II**: $w_1^{k+1,1} = \frac{1}{N} \sum_{l=2}^{N+1} w_1^{k,l}$
13: **Return** $w_1^{K+1,1}$

---

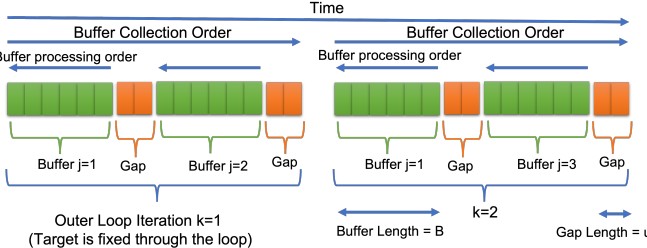

Figure 1: Illustration of Online Target Q-learning with Reverse Experience Replay

## 3 OUR ALGORITHM

As discussed in the introduction, we incorporate RER and OTL into Q-learning and introduce the algorithms Q-Rex (Online Target Q-learning with reverse experience replay, Algorithm 1), its sample efficient variant Q-RexDaRe (Q-Rex + data reuse, Algorithm 2) and its episodic variant EpiQ-Rex (Episodic Q-Rex, Algorithm 3). Since Q-RexDaRe and EpiQ-Rex are only minor modifications of Q-Rex, we refer the reader to the appendix for their pseudocode.

Q-Rex is parametrized by $K$ the number of iterations in the outer-loop, $N$ the number of buffers within an outer-loop iteration, $B$ the size of a buffer and $u$ the gap between the buffers. The algorithm has a three-loop structure where at the start of every outer-loop iteration (indexed by $k \in [K]$), we checkpoint our current guess of the $Q$ function given by $w_1^{k,1}$. Each outer-loop iteration corresponds to an inner-loop over the buffer collection with $N$ buffers, i.e. at iteration $j \in [N]$, we collect a buffer of size $B + u$ consecutive state-action-reward tuples. For every collected buffer we consider the first $B$ collected experiences and perform the target based Q-learning update in the reverse order for these experiences. We refer Figure 1 for an illustration of the processing order. Of note, is the usage of checkpointed target network in the RHS of the Q-learning update through the entirety of the outer-loop iteration, i.e. for a fixed $k$ and for all $j, i$, our algorithm sets

$$w_{i+1}^{k,j} = w_i^{k,j} + \eta \left[ r_{B+1-i}^{k,j} + \gamma \max_{a' \in \mathcal{A}} \langle \phi(s_{B+2-i}^{k,j}, a'), \mathbf{w_1^{k,1}} \rangle - \langle \phi_{B+1-i}^{k,j}, w_i^{k,j} \rangle \right] \phi_{B+1-i}^{k,j}$$

Figure 1 provides an illustration of the processing order for our updates. It can be seen that the number of experiences collected through the run of the algorithm is $T = KN(B+u)$. For the sake of simplicity, we will assume that the initial point, $w_1^{1,1} = 0$. Essentially the same results hold for arbitrary initial conditions. Q-RexDaRe is a modification of Q-Rex where we re-use the data from the first outer-loop iteration (i.e, data from $k = 1$) in every outer-loop iteration (i.e, $k > 1$).

| Setting | $K$ | $N$ | $u$ | $B$ | $\eta$ |
|---|---|---|---|---|---|
| ZIBEL MDP (THEOREM 1) | $\geq 1$ | $> \frac{C_3}{B}\frac{\kappa}{\eta}\log\left(\frac{K\kappa}{\delta(1-\gamma)}\right)$ | $\geq C_1\tau_{\mathsf{mix}}\log(\frac{C_{\mathsf{mix}}KN}{\delta})$ | $=10u$ | $< C_2\min(\frac{(1-\gamma)^2}{C_\Phi^2+\log(K/\delta)},\frac{1}{B})$ |
| TABULAR MDP (THEOREM 2) | $\geq C_2\frac{\left(\log\left(\frac{1}{1-\gamma}\right)\right)^2}{1-\gamma}$ | $> \frac{C_4}{B}\frac{\tau_{\mathsf{mix}}}{\mu_{\min}}\log(\frac{|S||A|K}{\delta})$ | $\geq C_1\tau_{\mathsf{mix}}\log(\frac{KN}{\delta})$ | $=10u$ | $< \frac{C_3}{\log\frac{|\mathcal{S}||\mathcal{A}|K}{\delta}}$ |
| TABULAR MDP (THEOREM 3) | $\geq C_2\frac{\log\left(\frac{1}{1-\gamma}\right)}{1-\gamma}$ | $> \frac{C_4}{B}\frac{\tau_{\mathsf{mix}}}{\mu_{\min}}\log(\frac{|S||A|}{\delta})$ | $\geq C_1\tau_{\mathsf{mix}}\log(\frac{KN}{\delta})$ | $=10u$ | $< C_3\frac{(1-\gamma)^2}{\bar{d}\log\left(\frac{|\mathcal{S}||\mathcal{A}|}{\delta}\right)}$ |
| ZIBEL MDP (THEOREM 1) | $\frac{\beta_1}{(1-\gamma)}$ | $\kappa\beta_2\max\left(\frac{C_\Phi^2+\beta_2}{\epsilon^2(1-\gamma)^4\tau_{\mathsf{mix}}},1\right)$ | $\tau_{\mathsf{mix}}\log\left(\frac{KN}{\delta}\right)$ | $10u$ | $\min\left(\frac{(1-\gamma)^4\epsilon^2}{C_\Phi^2+\beta_3},\frac{1}{B}\right)$ |
| TABULAR MDP (THEOREM 2) | $\frac{\beta_1^2}{1-\gamma}$ | $\frac{1}{\mu_{\min}}\max\left(\beta_5,\frac{\beta_1}{\eta\tau_{\mathsf{mix}}}\right)$ | $\tau_{\mathsf{mix}}\log\left(\frac{KN}{\delta}\right)$ | $10u$ | $\frac{(1-\gamma)^3}{\beta_5}\min(\epsilon,\epsilon^2)$ |
| TABULAR MDP (THEOREM 3) | $\frac{\beta_1}{(1-\gamma)}$ | $\frac{1}{\mu_{\min}}\max\left(\beta_5,\frac{\beta_1}{\eta\tau_{\mathsf{mix}}}\right)$ | $\tau_{\mathsf{mix}}\log\left(\frac{KN}{\delta}\right)$ | $10u$ | $\min\left(\frac{\epsilon^2(1-\gamma)^3}{\beta_4},\frac{(1-\gamma)^2}{d\beta_5},\frac{\epsilon(1-\gamma)^3}{\sqrt{d}\beta_4}\right)$ |

Table 2: Parameter constraints (first 3 rows) and choice for $< \epsilon$ error (last 3 rows) for our algorithms. Here the poly-log factors $\beta_i$ are given by $\beta_1 = \log\left(\frac{1}{(1-\gamma)\min(\epsilon,1)}\right)$, $\beta_3 = \log\left(\frac{1}{(1-\gamma)\delta\min(\epsilon,1)}\right)$, $\beta_2 = \log(\kappa) + \beta_3$, $\beta_4 = \log\left(\frac{|\mathcal{S}||\mathcal{A}|K}{\delta}\right)$, $\beta_5 = \log\left(\frac{|\mathcal{S}||\mathcal{A}|}{\delta}\right)$.

**Remark 2.** *For the sake of clarity, we only analyze the algorithms Q-Rex and Q-RexDaRe for data from a single trajectory with **Option I**. Option II involves averaging of the iterates which boosts the convergence – indeed we can obtain much better bounds in this setting by using standard analysis.*

## 4 MAIN RESULTS

We will now provide finite time convergence analysis and sample complexity for the algorithms Q-Rex and Q-RexDaRe. Recall that $K$ is the number of outer-loops, $N$ is the number of buffers inside an outer-loop iteration, $B$ is the buffer size and $u$ is the size of the gap. In what follows, we will take $u = \tilde{O}(\tau_{\mathsf{mix}})$, $B = 10u$, $K = \tilde{O}(\frac{1}{1-\gamma})$. We also note that the total number of samples used is $NK(B + u)$ for Q-Rex and $N(B + u)$ for Q-RexDaRe since we reuse data in each outer-loop iteration. In what follows, by $Q_1^{K+1,1}(s, a)$, we denote $\langle\phi(s, a), w_1^{K+1,1}\rangle$ which is our estimate for the optimal $Q$ function. Here $w_1^{K+1,1}$ is the output of either Q-Rex or Q-RexDaRe at the end of $K$ outer-loop iterations. Define $\|Q_1^{K+1,1} - Q\|_\infty := \sup_{(s,a)\in\mathcal{S}\times\mathcal{A}}\left|Q_1^{K+1,1}(s, a) - Q^*(s, a)\right|$. We first consider the performance of Q-Rex with data derived from a linear MDP (Defintion 1) or a ZIBEL MDP (Definition 2) and satisfying the Assumptions in Section 2.1.

**Theorem 1** (ZIBEL /Linear MDP). *Suppose we run Q-Rex using **Option I** with data from an MDP with $\mathsf{IBE} = 0$. There exists constants $C_1, C_2, C_3, C_4, C_5 > 0$ such that whenever the parameter bounds given in Table 2 (row 1) are satisfied, then with probability at-least $1 - \delta$, we must have:*

$$\|Q_1^{K+1,1} - Q^*\|_\infty \leq \frac{\gamma^K}{1-\gamma} + C_4\sqrt{\frac{K\kappa}{\delta(1-\gamma)^4}}\exp\left(-\frac{\eta NB}{\kappa}\right) + C_5\sqrt{\frac{\eta\left[C_\Phi^2+\log\left(\frac{K}{\delta}\right)\right]}{(1-\gamma)^4}}$$

*Given $\epsilon \in (0, \frac{1}{(1-\gamma)}]$, and the parameters as given in Table 2 (row 4)(up to constant factors), then with probability at-least $1 - \delta$: $\|Q_1^{K+1,1}(s, a) - Q^*(s, a)\|_\infty < \epsilon$. This has a sample complexity*

$$\Theta(NKB) = \tilde{O}\left(\kappa\max\left(\frac{C_\Phi^2+1}{(1-\gamma)^5\epsilon^2}, \frac{\tau_{\mathsf{mix}}}{1-\gamma}\right)\right)$$

We now consider the performance of Q-Rex and Q-RexDaRe in the case of tabular MDPs. We refer to Table 1 for a comparison of our results to the state-of-art results provided in literature for Q-learning based algorithms.

**Remark 3.** *To the best of our knowledge, Theorem 1 presents the first non-asymptotic convergence results for Q-learning based methods for ZIBEL MDPs under standard assumptions. Notice that the sample complexity scales as $\frac{1}{\epsilon^2} + \tau_{\mathsf{mix}}$ instead of $\frac{\tau_{\mathsf{mix}}}{\epsilon^2}$ like in Zou et al. (2019b); Bhandari et al. (2018). This is because in the case of ZIBEL MDPs RER brings out the super-martingale structure present in the problem which forward pass does not.*

**Theorem 2** (Tabular MDP). *Suppose we run Q-Rex using **Option I** with data derived from tabular MDPs. Whenever the algorithmic parameters are picked as given in Table 2 (row 2) for some universal constants $C_1, \ldots, C_5$, we obtain with probability at-least $1 - \delta$:*

$$\|Q^{K+1} - Q^*\|_\infty < C_5\left[\frac{\gamma^L}{1-\gamma} + \frac{\exp\left(-\frac{\eta\mu_{\min}NB}{2}\right)}{(1-\gamma)^2} + \frac{\eta\log\left(\frac{K|\mathcal{S}||\mathcal{A}|}{\delta}\right)}{(1-\gamma)^3} + \sqrt{\frac{\eta\log\left(\frac{K|\mathcal{S}||\mathcal{A}|}{\delta}\right)}{(1-\gamma)^3}}\right.$$

*Where $L = \frac{b_1 K}{\log \frac{1}{1-\gamma}}$. Given $\epsilon \in (0, \frac{1}{1-\gamma}]$, and the parameters are picked as given in Table 2 (row 5), then with probability at-least $1 - \delta$, we have: $\|Q^{K+1} - Q^*\|_\infty < \epsilon$. This gives us a sample complexity of*

$$\Theta(NKB) = \tilde{O}\left(\frac{1}{\mu_{\min}} \max\left(\frac{1}{(1-\gamma)^4 \min(\epsilon, \epsilon^2)}, \frac{\tau_{\text{mix}}}{1-\gamma}\right)\right).$$

Even though the sample complexity provided in Theorem 2 matches the sample complexity of *synchronous* Q-learning even with *asynchronous* data, it is still sub-optimal with respect to the min-max lower bounds (i.e, it has a dependence of has a dependence of $\frac{1}{(1-\gamma)^4}$ instead of the optimal $\frac{1}{(1-\gamma)^3}$). We resolve this gap for Q-RexDaRe in Theorem 3. For tabular MDPs, the number states can be large but the support of $P(\cdot|s, a)$ is bounded in most problems of practical interest. Consider the following assumption (note that this holds for every tabular MDP with $\bar{d} = |\mathcal{S}|$.)

**Assumption 5.** *Tabular MDP is such that $|\mathsf{supp}(P(\cdot|s, a))| \leq \bar{d} \in \mathbb{N}$.*

**Theorem 3** (Tabular MDP with Data Reuse). *For tabular MDPs, suppose additionally Assumption 5 holds and we run Q-RexDaRe using **Option I**. There exist universal constants $C_1, C_2, C_3, C_4$ such that when the parameter values satisfy the bounds in Table 2 (row 3), with probability at-least $1 - \delta$:*

$$\|Q_1^{K+1,1} - Q^*\|_\infty \leq C\left[\frac{\exp\left(-\frac{\eta\mu_{\min}NB}{2}\right) + \gamma^K}{(1-\gamma)^2} + \frac{\eta \log\left(\frac{|\mathcal{S}||\mathcal{A}|K}{\delta}\right)}{(1-\gamma)^3}\sqrt{\bar{d}} + \sqrt{\frac{\eta}{(1-\gamma)^3}\log\left(\frac{K|\mathcal{S}||\mathcal{A}|}{\delta}\right)}\right]$$

*Suppose $\epsilon \in (0, \frac{1}{1-\gamma}]$. If we choose the parameters as per Table 2 (row 6), then with probability at-least $1 - \delta$ we have: $\|Q^{K+1}(s, a) - Q^*\|_\infty < \epsilon$. The sample complexity in this case is*

$$\Theta(NB) = \tilde{O}\left(\frac{1}{\mu_{min}} \max\left(\tau_{\text{mix}}, \frac{1}{\epsilon^2(1-\gamma)^3}, \frac{\bar{d}}{(1-\gamma)^2}, \frac{\sqrt{\bar{d}}}{\epsilon(1-\gamma)^3}\right)\right).$$

## 5 OVERVIEW OF THE ANALYSIS

We divide the analysis of Q-Rex and Q-RexDaRe into two parts: Analysis of $w_1^{k,1}$ obtained at the end of outer-loop iteration $k$ and the analysis of the algorithm within the outer-loop. The algorithm reduces to SGD for linear regression with Markovian data within an outer-loop due to OTL. That is, we try to find $w_1^{k+1,1}$ such that $\langle w_1^{k+1,1}, \phi(s, a)\rangle \approx R(s, a) + \mathbb{E}_{s' \sim P(\cdot|s,a)} \sup_{a'}\langle w_1^{k,1}, \phi(s', a')\rangle$. Therefore, we write $w_1^{k+1,1} = \mathcal{T}(w_1^{k,1}) + \epsilon_k(w_1^{k,1})$, where $\mathcal{T}$ is the $\gamma$ contractive Bellman operator whose unique fixed point is $w^*$ and $\epsilon_k$ is the noise to be controlled. Following a similar setting in in (Jain et al., 2021b), we control $\epsilon_k$ with the following steps:
(1) We introduce a fictitious coupled process (see Section C) $(\tilde{s}_t, \tilde{a}_t, \tilde{r}_t)$ where the data in different buffers are *exactly* independent (since the gaps of size $u$ make the buffers *approximately* independent) and show that the algorithm run with the fictitious data has the same output as the algorithm run with the actual data with high probability when $u$ is large enough.
(2) We give a bias-variance decomposition (Lemma 5) for the error $\epsilon_k$ where the exponentially decaying bias term helps forget the initial condition and the variance term arises due the inherent noise in the samples.
(3) We control the bias and variance terms separately in order to ensure that the noise $\epsilon_k$ is small enough. RER plays a key role in controlling the variance term by endowing it with a super-martingale structure, which is not possible with forward order traversal (see Theorem 4).

The procedure described above allows us to show that $w_1^{k+1,1} \approx \mathcal{T}(w_1^{k,1})$ uniformly for $k \leq K$, which directly gives us a convergence bound to the fixed point of $\mathcal{T}$ i.e, $w^*$ (Theorem 1). In the tabular case, the approximate Bellman iteration connects to the analysis of *synchronous* Q-learning in (Li et al., 2021), which allows us to obtain a better convergence guarantee (Theorem 2). To obtain convergence guarantees for Q-RexDaRe, we first observe that if we re-use the data used in outer-loop iteration 1 in all future outer-loop iterations $k > 1$, $\epsilon_k(w_1^{k,1})$ might not be small since $w_1^{k,1}$ depends on $\epsilon_k(\cdot)$. However, $(w_1^{k,1})_k$ approximates the deterministic path of the noiseless Bellman iterates: $\bar{w}_1^{1,1} := w_1^{1,1}$ and $\bar{w}_1^{k+1,1} := \mathcal{T}(\bar{w}_1^{k,1})$. Since $\|\epsilon_k(w_1^{k,1})\|_\infty \leq \|\epsilon^k(w_1^{k,1}) - \epsilon^k(\bar{w}_1^{k,1})\|_\infty + \|\epsilon^k(\bar{w}_1^{k,1})\|_\infty$, we argue inductively that $\|\epsilon^k(w_1^{k,1}) - \epsilon^k(\bar{w}_1^{k,1})\|_\infty \approx 0$ since $w_1^{k,1} \approx \bar{w}_1^{k,1}$ and $\|\epsilon^k(\bar{w}_1^{k,1})\|_\infty \approx 0$ since $\bar{w}_1^{k,1}$ is a deterministic sequence and hence $w_1^{k+1,1} \approx \bar{w}_1^{k+1,1}$.

ACKNOWLEDGMENTS

Most of this work was done when D.N. was a graduate student at MIT and was supported in part by NSF grant DMS-2022448. Part of this work was done when D.N. was a visitor at the Simons Institute for Theory of Computing, Berkeley. We would also like to thank Gaurav Mahajan for introducing us to low-inherent Bellman error setting, and providing intuition that our technique might be applicable in this more general setting (than linear MDP) as well.

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

# A  EXPERIMENTS

Even though OTL has a stabilizing effect on the Q-iteration, it reduces the rate of bias decay since the values are not updated for a long time. Therefore, the success of our procedure depends on picking the right values for the parameters $N, B$ and **Option I** vs. **Option II**. However, under the right conditions the algorithms which include OTL+RER converge to a much smaller final error as illustrated by the examples we provide in this section. If a better sample complexity is desired, then Q-RexDaRe can be used as shown by Theorem 3. Further research is needed to identify the practical conditions such as (function approximation coarseness, MDP reward structure etc.) under which techniques like OTL+RER help.

**100-State Random Walk**  We first consider the episodic MDP (Sutton & Barto, 2018, example 6.2) – but with 100 states instead of 1000. Here the agent can either move left or right on a straight line, receiving a reward of 0 at each step. Reaching the right terminal point ends the episode with reward 1, while the left terminal point ends the episode with reward 0. In each episode, the initial point is uniformly random and the offline policy chooses right and left directions uniformly at random. We use state aggregation to obtain a linear function representation (Sutton & Barto, 2018) with total 10 aggregate states. Along with 2 actions, this leads to a 20 dimensional embedding. We compare vanilla Q-learning, OTL+ER+Q-learning and EpiQ-Rex (**Option II**, $N = 1$). The same step size (0.01) was chosen for all the algorithms. OTL+ER+Q has the same structure as EpiQ-Rex and was run with the same parameters as EpiQ-Rex . The main difference between the two algorithms is that OTL+ER+Q processes the experiences collected in each episode in a random order processing instead of reverse order. Refering to Figure 4, we note that the bias decay for EpiQ-Rex and OTL+ER+Q are slower than vanilla Q-learning due to the online target structure. However EpiQ-Rex converges to a better solution than the other algorithms.

**Mountain Car**  We run an online control type experiment with the Mountain car problem (Sutton & Barto, 2018, Example 10.1). The task here is to control the car and help it reach the correct peak (which ends the episode) as soon as possible (i.e, terminate the episode with fewest steps). The agent receives a reward of $-1$ unless the correct peak is reached. Here we run EpiQ-Rex with **Option I** and $N = 1$, OTL+ER+Q-learning and vanilla Q-learning. The $k$-th episode is generated with the policy at the end of $k - 1$-th episode for each of the three algorithms. We use a $n = 4$ tile coding to represent the Q values for a given action, each of which has $4 \times 4$ squares. The step-size was picked as $0.1/n$ and the result was averaged over 500 runs of the experiment. We refer to Figure 2 for the outcomes. For the last 300 episodes, the mean episode length for Vanilla Q-learning was 145, EpiQ-Rex was 136 and OTL+ER+Q-learning was 143 (rounded to the nearest integer).

**Grid-World**  We consider the grid-world problem which is a tabular MDP (Sutton & Barto, 2018, Example 3.5), which is a continuing task. Here, an agent can walk north, south, east or west in a $5 \times 5$ grid. Trying to fall off the grid accrues a reward of $-1$, while reaching certain special states accrues a reward of 10 or 5. In our example, we also add a unif$[-0.5, 0.5]$ noise to all rewards to make the problem harder. We run the grid world experiment with the discount factor $\gamma = 0.9$ and step size 0.05 for Vanilla Q-learning, Q-Rex (**Option II**, $B = 3000$, $N = 1$) and OTL+ER+Q-learning which is run with the same parameters as Q-Rex but with random samples from the buffer. We run the experiment 30 times and plot the error in the Q-values vs. time in Figure 3.

**Baird's Counter-Example**  Consider the famous Baird's counter-example shown in Figure 6. The features corresponding to each state is shown and the reward for any transition is 0. Thus, $w^* = 0$ is the optimal solution. Since Assumption 4 made in this work is violated for this example, we consider the analog of the problem where at each step, a state is sampled uniformly at random and the corresponding transition, along with the 0 reward is used to learn the vector $w$. In the experiment, we set the problem and algorithmic parameters to be $\gamma = 0.99$ and $\eta = 0.01/\sqrt{5}$ (the factor of $\sqrt{5}$ normalizes the features to satisfy Assumption 1). We set $B = 50$, $u = 0$, and $N = 5$ since there is no need for keeping a gap between the buffers in this experiment. Figure 7 shows the non-convergent behavior of vanilla Q-learning while OTL-based Q-learning converges. Note that since the sampling of state, action and reward is done in a uniformly random fashion, it is not relevant to use reverse experience replay. It is easy to see that with data reuse, we only need few samples to ensure all states are covered; the rate of convergence would be same as that of OTL+Q-learning.

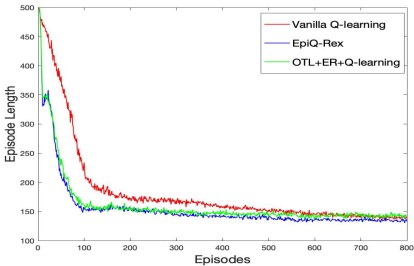

Figure 2: Mountain Car problem

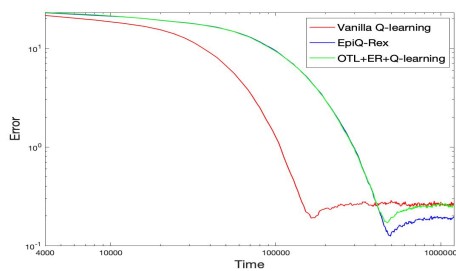

Figure 3: Grid-world problem

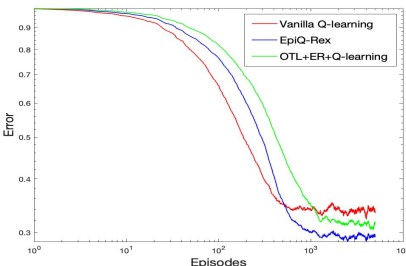

Figure 4: 100 State Random Walk

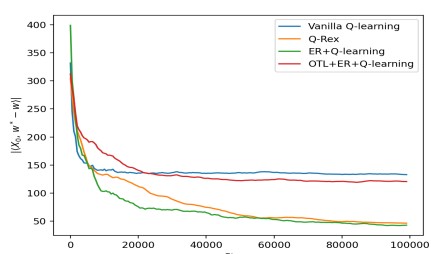

Figure 5: Linear System Problem

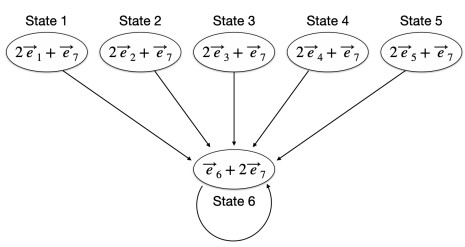

Figure 6: Feature embedding in the modified Baird's counter-example; $\vec{e_i}$ represents the $i$-th canonical basis vector in $\mathbb{R}^7$. Each transition shown occurs with probability 1.

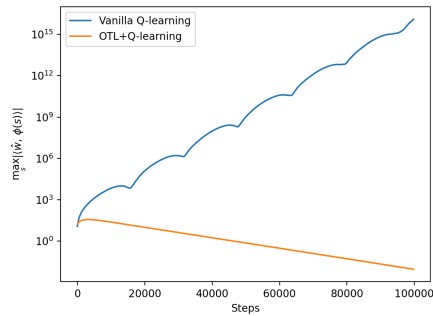

Figure 7: Performance of vanilla Q-learning as compared to OTL+Q-learning, averaged over 10 independent runs

**Linear Dynamical System with Linear Rewards**   We also compare the algorithms on a linear dynamical systems problem. While the problem described next is strictly not a linear MDP, the value functions for certain policies can be written as a linearly in terms of the initial condition; this is made precise next. Consider a linear dynamical system with initial state being $X_0 \in \mathbb{R}^d$. The state evolves as $X_{t+1} = AX_t + \eta_t$, where $A \in \mathbb{R}^{d \times d}$. The reward obtained for such transition is given by $\langle X_t, \theta \rangle$ for a fixed $\theta \in \mathbb{R}^d$. The maximum singular value of $A$ is chosen less than 1 to ensure that the system is stable. The infinite horizon $\gamma$-discounted expected reward is given by $\langle X_0, \sum_{i=0}^{\infty} (\gamma A^T)^i \theta \rangle$. Thus, the expected reward can be written as $\langle X_0, w^* \rangle$, where

$$w^* = (I - \gamma A^T)^{-1} \theta.$$

Since there are no actions in this case, the Q learning algorithms reduce to value function approximation (i.e, TD(0) type algorithms). We take the embedding $\phi(X_t) = X_t$, the identity mapping. We considered $d = 5$, $\gamma = 0.99$, $\eta = 0.01$, and a randomly generated normal matrix $A$ and $\theta$. We Option II for Q-Rex along with $B = 75$, $u = 25$, and $N = 5$ for the experiments. For OTL+ER+Q-learning, we keep the same parameters, but with random order sampling, while ER+Q-learning does not include OTL. The results shown in Figure 5 are averaged over 100 independent runs of the experiment. Note that the errors considered are using iterate averaging.

As seen from Figure 5, Q-Rex outperforms vanilla Q-learning and OTL+ER-based Q-learning as one would expect based on the theory presented in this work. However, it is interesting to note that ER+Q-learning performs slightly better than Q-Rex. One possible reason could be due to the fact that in Q-Rex, the target gets updated at a slower rate at the cost of reducing bias. However, setting a smaller value of $NB$ might resolve the issue.

# B   DEFINITIONS AND NOTATIONS

## B.1   Q-REXDARE

The psuedocode for Q-RexDaRe is given in Algorithm 2. Note that we reuse the sample data in every outer-loop iteration instead of drawing fresh samples.

---

**Algorithm 2** Q-RexDaRe

1: **Input:** learning rates $\eta$, horizon $T$, discount factor $\gamma$, trajectory $X_t = \{s_t, a_t, r_t\}$, number of outer-loops $K$ buffer size $B$, buffer gap $u$
2: Total buffer size: $S \leftarrow B + u$
3: Number of buffers loops: $N \leftarrow T/S$
4: $w_1^{1,1} = 0$ initialization
5: **for** $k = 1, \ldots, K$ **do**
6:      **for** $j = 1, \ldots, N$ **do**
7:          Form buffer $\mathsf{Buf} = \{X_1^{k,j}, \ldots, X_S^{k,j}\}$, where,

$$X_i^{k,j} \leftarrow X_{S(j-1)+i}$$

8:          Define for all $i \in [1, S]$, $\phi_i^{k,j} \triangleq \phi(s_i^{k,j}, a_i^{k,j})$.
9:          **for** $i = 1, \ldots, B$ **do**
10:             $w_{i+1}^{k,j} = w_i^{k,j} + \eta \left[ r_{B+1-i}^{k,j} + \gamma \max_{a' \in \mathcal{A}} \langle \phi(s_{B+2-i}^{k,j}, a'), w_1^{k,1} \rangle - \langle \phi_{B+1-i}^{k,j}, w_i^{k,j} \rangle \right] \phi_{B+1-i}^{k,j}$
11:          $w_1^{k,j+1} = w_{B+1}^{k,j}$
12:      $w_1^{k+1,1} = w_1^{k,N+1}$
13: **Return** $w_0^{T+1}$

---

## B.2   EPIQ-REX

The psuedocode for EpiQ-Rex is given in Algorithm 3. Note that the buffers here are individual episodes and can vary in size due to inherent randomness. We do not require a gap here since separate episodes are assumed to be independent.

**MDP definition**   Here we construct the MDP with a (possibly) random reward $r$. We consider non-episodic, i.e. infinite horizon, time homogenous Markov Decision Processes (MDPs) and we denote an MDP by $\mathrm{MDP}(\mathcal{S}, \mathcal{A}, \gamma, P, r)$ where $\mathcal{S}$ denotes the state space, $\mathcal{A}$ denotes the action space, $\gamma$ represents the discount factor, $P(s'|s, a)$ represents the probability of transition to state $s'$ from the state $s$ on action $a$. We assume for purely technical reasons that $\mathcal{S}$ and $\mathcal{A}$ are compact subsets of $\mathbb{R}^n$ for some $n \in \mathbb{N}$). $r$ is a reward process (not to be confused with Markov Reward Processes i.e, MRP) indexed by $\mathcal{S} \times \mathcal{A} \times \mathcal{S}$, such that $r(s, a, s') \in [0, 1]$ almost surely. We will skip the measure theoretic details of the definitions and assume that the sequence of i.i.d. reward processes $(r_t)_{t \in \mathbb{N}}$ can be jointly defined over a Polish probability space. The function $R : \mathcal{S} \times \mathcal{A} \to [0, 1]$ represents the deterministic reward $R(s, a)$ obtained on taking action $a$ at state $s$ and is defined by $R(s, a) = \mathbb{E} \left[ \mathbb{E}_{s' \sim P(\cdot|s,a)} r(s, a, s') \right]$. Here the expectation is with respect to both the randomness in the reward process and in state $s'$.

Now, given a trajectory, $(s_t, a_t)_{t=1}^T$, which is independent of i.i.d sequence of rewards processes $(r_t)_{t=1}^T$. We observe $(s_t, a_t, r_t(s_t, a_t, s_{t+1}))_{t=1}^T$, and we will henceforth denote $r_t(s_t, a_t, s_{t+1})$ by just $r_t$.

---

**Algorithm 3** EpiQ-Rex

---

1: **Input:** learning rates $\eta$, number of episodes $T$, discount factor $\gamma$, Epsiodes $E_t = \{(s_1^t, a_1^t, r_1^t), \ldots, (s_{B_t}^y, a_{B_t}^t, r_{B_t}^t)\}$, Number of buffer per outer-loop iteration N.

2: Number of outer loop iterations: $K \leftarrow T/N$

3: $w_1^{1,1} = 0$ initialization

4: **for** $k = 1, \ldots, K$ **do**

5:     **for** $j = 1, \ldots, N$ **do**

6:         $t \leftarrow (k-1) * N + j$

7:         Collect experience and form buffer $\mathsf{Buf} = \{X_1^t, \ldots, X_{B_t}^t\}$, where,

$$X_i^t = (s_i^t, a_i^t, r_i^t)$$

8:         Define for all $i \in [1, B_t - 1]$, $\phi_i^{k,j} \triangleq \phi(s_i^{k,j}, a_i^{k,j})$.

9:         **for** $i = 1, \ldots, B$ **do**

10:             $w_{i+1}^{k,j} = w_i^{k,j} + \eta \left[ r_{B_t-i}^{k,j} + \gamma \max_{a' \in \mathcal{A}} \langle \phi(s_{B_t+1-i}^{k,j}, a'), w_1^{k,1} \rangle - \langle \phi_{B_t-i}^{k,j}, w_i^{k,j} \rangle \right] \phi_{B_t-i}^{k,j}$

11:         **Option I:** $w_1^{k,j+1} = w_{B_t}^{k,j}$

12:         **Option II:** $w_1^{k,j+1} = \frac{1}{B_t-1} \sum_{i=2}^{B_t} w_i^{k,j}$

13:   $w_1^{k+1,1} = w_1^{k,N+1}$

14: **Return** $w_1^{K,1}$

---

**Notation** Due to three loop nature of our algorithm it will be convenient to define some simplifying notation. To this end, consider the outer loop with index $k \in [K]$ and buffer number $j \in [N]$ inside this outer loop. Further given an $i \in [S]$ define a time index as $t_i^{k,j} = NS(k-1) + S(j-1) + i$. We now denote the $i$-th state-action-reward tuple inside this buffer by

$$(s_i^{k,j}, a_i^{k,j}, r_i^{k,j}) = (s_{t_i^{k,j}}, a_{t_i^{k,j}}, r_{t_i^{k,j}}).$$

Similarly, $R_i^{k,j} = R(s_i^{k,j}, a_i^{k,j})$. For conciseness, we define for all $i, j, k$, $\phi_i^{k,j} := \phi(s_i^{k,j}, a_i^{k,j})$. Since we are processing the data in the reverse order within the buffer, the following notation will be useful for analysis:

$$s_{-i}^{k,j} := s_{B+1-i}^{k,j}, a_{-i}^{k,j} := a_{B+1-i}^{k,j}, r_{-i}^{k,j} := r_{B+1-i}^{k,j}, R_{-i}^{k,j} := R_{B+1-i}^{k,j} \text{ and } \phi_{-i}^{k,j} := \phi_{B+1-i}^{k,j}.$$

## C  COUPLED PROCESS

It can be seen that the buffers are approximately i.i.d. whenever we take $u = O(\tau_{\mathsf{mix}} \log \frac{T}{\delta})$ whenever Assumption 3 is satisfied. For the sake of clarity of analysis, we will consider exactly independent buffers. That is, we assume the algorithms are run with a fictitious trajectory $(\tilde{s}_t, \tilde{a}_t, \tilde{r}_t)$, where we assume that the fictitious trajectory is generated such that the first state of every buffer is sampled from the stationary distribution $\mu$. We show that we can couple this fictitious process (i.e, define it on a common probability space as the original process $(s_t, a_t, r_t)$) such that

$$\mathbb{P}\left( \cap_{k=1}^K \cap_{j=1}^N \cap_{i=1}^{B+1} \{(s_i^{k,j}, r_i^{k,j}, a_i^{k,j}) = (\tilde{s}_i^{k,j}, \tilde{r}_i^{k,j}, \tilde{a}_i^{k,j})\} \right) \geq 1 - \delta. \tag{4}$$

Notice that the equality does not hold within the gaps between the buffer which are of size $u$ but inside the buffers of size $B$ only. That is, the sequence of iterates obtained by running the algorithm with the original data $(s_t, a_t, r_t)$ is the same as the sequence of iterates obtained by running the algorithm with the fictitious coupled data $(\tilde{s}_t, \tilde{a}_t, \tilde{r}_t)$ with high probability. We state this result formally in Lemma 16 and prove it in Section M.1. Henceforth, we will assume that we run the algorithm with data $(\tilde{s}_t, \tilde{a}_t, \tilde{r}_t)$ and refer to Lemma 16 to carry over the results to the original data set with high probability. We analogously define $\tilde{\phi}_i^{k,j}$. We will denote the iterates of the algorithm run with the coupled trajectory as $\tilde{w}_i^{k,j}$ instead of $w_i^{k,j}$ and will focus on it entirely. We now provide some definitions based on the above process. These definitions will be used repeatedly in our analysis.

## D    BASIC STRUCTURAL LEMMAS

We first note some basic structural lemmas regarding ZIBEL MDPs under the assumptions in Section 2.1. We refer to Section M for the proofs.

**Lemma 1.** *For tabular MDPs satisfying the assumptions in Section 2.1, for both Q-Rex  and Q-RexDaRe, we have that for every $k, j, i$ we have that $\|\tilde{w}_i^{k,j}\|_\phi \leq \frac{1}{1-\gamma}$*

The lemma above says, in particular, that the Q-value estimate given by our algorithm never exceeds $\frac{1}{1-\gamma}$ due to 0 initialization. The proof is a straightforward induction argument, which we omit. We will henceforth use Lemma 1 without explicitly mentioning it.

**Lemma 2.** *Suppose $Q_w(s,a) := \langle w, \phi(s,a)\rangle$ and let $Q^*(s,a) = \langle w^*, \phi(s,a)\rangle$ be the optimal Q function. Then:*

$$\sup_{(s,a)\in\mathcal{S}\times\mathcal{A}} |Q_w(s,a) - Q^*(s,a)| = \|w - w^*\|_\phi$$

*Moreover, we must have: $\|x\| \geq \|x\|_\phi \geq \frac{\|x\|}{\sqrt{\kappa}}$ for any $x \in \mathbb{R}^d$.*

**Lemma 3.** *For any $w_0 \in \mathbb{R}^d$, there exists a unique $w_1 \in \mathbb{R}^d$ such that*

$$\langle w_1, \phi(s,a)\rangle = R(s,a) + \gamma\mathbb{E}_{s'\sim P(\cdot|s,a)} \sup_{a'\in A} \langle \phi(s',a'), w_0\rangle.$$

*We will denote this mapping $w_0 \to w_1$ by $w_1 = \mathcal{T}(w_0)$.*

**Lemma 4.** *$\mathcal{T} : \mathbb{R}^d \to \mathbb{R}^d$ is $\gamma$ contractive in the norm $\|\cdot\|_\phi$. The unique fixed point of $\mathcal{T}$ is $w^*$. Moreover, we have: $\|w^*\|_\phi \leq \frac{1}{1-\gamma}$ and $\|w^*\| \leq \frac{\sqrt{\kappa}}{1-\gamma}$.*

In view of Lemma 4, we can begin to look at the following *noiseless* Q-iteration. Let $\bar{w}^1 = w_1^{1,1} = 0$ and $\bar{w}^{k+1} = \mathcal{T}(\bar{w}^k)$. This converges geometrically to $w^*$ with contraction coefficient $\gamma$ under the norm $\|\cdot\|_\phi$. In our case, however, we only have sample access to the operator $\mathcal{T}$. Therefore, our Q-iteration at the end of $k$-th outer loop can be written as $\tilde{w}_1^{k+1,1} = \mathcal{T}(\tilde{w}_1^{k,1}) + \tilde{\epsilon}_k$ where $\tilde{\epsilon}_k$ is the error introduced via sampling which needs to be controlled.

## E    BIAS VARIANCE DECOMPOSITION

We begin our analysis by providing a bias-variance decomposition of the error with respect to the noiseless Q-iteration at every loop. We will need the following definitions which we use repeatedly through our analysis.

Given step size $\eta$, outer loop index $k$ and buffer index $j$, we define the following contraction matrices for $a, b \in [B]$.

$$\tilde{H}_{a,b}^{k,j} := \prod_{i=a}^{b} \left( I - \eta\tilde{\phi}_i^{k,j}[\tilde{\phi}_i^{k,j}]^\top \right) . \tag{5}$$

Whenever $a > b$, we will define $\tilde{H}_{a,b}^{k,j} := I$. In the tabular setting, we define for any $(s,a) \in \mathcal{S} \times \mathcal{A}$ by $\tilde{N}^k(s,a)$ to be the number of samples of $(s,a)$ seen in the outer loop k (excluding the gaps),i.e.

$$\tilde{N}^k(s,a) = \sum_{j=1}^{N} \sum_{i=1}^{B} \mathbb{1}((\tilde{s}_i^{k,j}, \tilde{a}_i^{k,j}) = (s,a))$$

Further we denote by $\tilde{N}_i^{k,j}(s,a)$, the number of samples of $(s,a)$ seen in the outer loop $k$ in the buffers post the buffer $j$ as well as the number of samples of $(s,a)$ in buffer $j$ before iteration $i$. Formally,

$$\tilde{N}_i^{k,j}(s,a) := \sum_{r=1}^{i-1} \mathbb{1}\left((\tilde{s}_r^{k,j}, \tilde{a}_r^{k,j}) = (s,a)\right) + \sum_{l=j+1}^{N} \sum_{r=1}^{B} \mathbb{1}\left((\tilde{s}_r^{k,l}, \tilde{a}_r^{k,l}) = (s,a)\right) .$$

We define the *error* term for any $w$:

$$\tilde{\epsilon}_i^{k,j}(w) := \left[ \tilde{r}_i^{k,j} - \tilde{R}_i^{k,j} + \gamma \sup_{a' \in \mathcal{A}} \langle w, \phi(\tilde{s}_{i+1}^{k,j}, a') \rangle - \gamma \mathbb{E}_{s' \sim P(\cdot | \tilde{s}_i^{k,j}, \tilde{a}_i^{k,j})} \sup_{a' \in A} \langle \phi(s', a'), w \rangle \right] . \quad (6)$$

Finally we define the following shorthands for any $i, j, k$:

$$\tilde{w}^{k+1,*} := \mathcal{T}(\tilde{w}_1^{k,1}) \qquad \tilde{\epsilon}_i^{k,j} := \tilde{\epsilon}_i^{k,j}(\tilde{w}_1^{k,1}) \qquad \tilde{L}^{k,j} := \eta \sum_{i=1}^{B} \tilde{\epsilon}_i^{k,j} \tilde{H}_{1,i-1}^{k,j} \tilde{\phi}_i^{k,j} .$$

Given the above definition, the following the lemma provides the Bias-Variance decomposition which is the core of our analysis.

**Lemma 5** (Bias-Variance Decomposition). *For every $k$, we have that,*

$$\tilde{\epsilon}_k := \tilde{w}_1^{k+1,1} - \tilde{w}^{k+1,*} = \prod_{j=N}^{1} \tilde{H}_{1,B}^{k,j} (\tilde{w}_1^{k,1} - \tilde{w}^{k+1,*}) + \sum_{j=1}^{N} \prod_{l=N}^{j+1} \tilde{H}_{1,B}^{k,l} \tilde{L}^{k,j} \quad (7)$$

Here and later on in the paper, we use the reverse order in the product to highlight the convention that higher indices $l$ in $\tilde{H}_{1,B}^{k,l}$ appear towards the left side of the product and further define $\prod_{l=N}^{N+1} \tilde{H}_{1,B}^{k,l} = I$. We call the first term in Equation (7) as the bias term and it decays geometrically with $N$ and the second term is called the variance, which has zero mean. We will bound these terms separately. Since tabular setting allows for improved analysis of error, we will provide special cases for the tabular setting with refined bounds. We refer to Section M.5 for the proof of Lemma 5.

# F    BOUNDING THE BIAS TERM

## F.1    TABULAR CASE

In the tabular case, we have the following expression for bias. We omit the proof since it follows from a simple calculation.

**Lemma 6.** *In the tabular setting, we have:*

$$\langle \phi(s,a), \prod_{j=N}^{1} \tilde{H}_{1,B}^{k,j} (\tilde{w}_1^{k,1} - \tilde{w}^{k+1,*}) \rangle = (1 - \eta)^{\tilde{N}^k(s,a)} \langle \tilde{w}_1^{k,1} - \tilde{w}^{k+1,*}, \phi(s,a) \rangle$$

For a particular outer loop $k$, we show that $\tilde{N}^k(s,a)$ is $\Omega(\mu_{\min} NB)$ with high probability whenever $NB$ is large enough. For this we use (Paulin, 2015, Theorem 3.4) similar to the proof of (Li et al., 2020b, Lemma 8). We give the proof in Section M.6.

**Lemma 7.** *There exists a constant $C$ such that whenever $B \geq \tau_{\mathsf{mix}}$ and $NB \geq C \frac{\tau_{\mathsf{mix}}}{\mu_{\min}} \log(\frac{|\mathcal{S}||\mathcal{A}|}{\delta})$, with probability at least $1 - \delta$, we have that for every $(s,a) \in \mathcal{S} \times \mathcal{A}$:*
$$\tilde{N}^k(s,a) \geq \tfrac{1}{2} \mu(s,a) NB \geq \tfrac{1}{2} \mu_{\min} NB$$

## F.2    ZIBEL MDP CASE

**Lemma 8.** *Suppose $g \in \mathbb{R}^d$ is fixed and $\eta B < \frac{1}{4}$. Then, the following hold:*

*1.*

$$\mathbb{E} \| \prod_{j=N}^{1} \tilde{H}_{1,B}^{k,j} g \|^2 \leq \exp(-\tfrac{\eta NB}{\kappa}) \| g \|^2 \quad (8)$$

*2. With probability at-least $1 - \delta$, we have:*

$$\mathbb{E} \| \prod_{j=N}^{1} \tilde{H}_{1,B}^{k,j} g \|_\phi \leq \exp(-\tfrac{\eta NB}{\kappa}) \sqrt{\tfrac{\kappa}{\delta}} \| g \|_\phi$$

We refer to Section L.1 for the proof.

## G   BOUNDING THE VARIANCE TERM

### G.1   TABULAR CASE

In the tabular case, it is clear that:

$$\langle \phi(s,a), \sum_{j=1}^{N} \prod_{l=N}^{j+1} \tilde{H}_{1,B}^{k,l} \tilde{L}^{k,j} \rangle = \sum_{j=1}^{N} \sum_{i=1}^{B} (1-\eta)^{\tilde{N}_i^{k,j}(s,a)} \eta \tilde{\epsilon}_i^{k,j} \mathbb{1}((\tilde{s}_i^{k,j}, \tilde{a}_i^{k,j}) = (s,a)) \qquad (9)$$

Now, we note that $\tilde{N}_i^{k,j}(s,a)$ depends on data in buffers $l > j$ and when $i_1 < i$ inside buffer $j$. Following the discussion in (Li et al., 2021), we define the vector $\mathsf{Var}_P(V_k), \mathsf{Var}_P(V^*) \in \mathbb{R}^{\mathcal{S} \times \mathcal{A}}$ such that

$$\mathsf{Var}_P(V_k)(s,a) = \mathbb{E}_{s' \sim P(\cdot|s,a)} \left[ \sup_{a' \in A} \langle \phi(s',a'), \tilde{w}_1^{k,1} \rangle - \mathbb{E}_{s' \sim P(\cdot|s,a)} \sup_{a' \in A} \langle \phi(s',a'), \tilde{w}_1^{k,1} \rangle \right]^2 \qquad (10)$$

$$\mathsf{Var}_P(V^*)(s,a) = \mathbb{E}_{s' \sim P(\cdot|s,a)} \left[ \sup_{a' \in A} \langle \phi(s',a'), w^* \rangle - \mathbb{E}_{s' \sim P(\cdot|s,a)} \sup_{a' \in A} \langle \phi(s',a'), w^* \rangle \right]^2 \qquad (11)$$

More generally, we define

$$\mathsf{Var}_P(V)(s,a;w) = \mathbb{E}_{s' \sim P(\cdot|s,a)} \left[ \sup_{a' \in A} \langle \phi(s',a'), w \rangle - \mathbb{E}_{s' \sim P(\cdot|s,a)} \sup_{a' \in A} \langle \phi(s',a'), w \rangle \right]^2$$

Similarly, we define $\tilde{L}^{k,j}(w)$ by replacing $w_1^{k,1}$ with any fixed, arbitrary $w$. It is easy to see that:

$$\mathbb{E}\left[ |\tilde{\epsilon}_i^{k,j}(w)|^2 \Big| (\tilde{s}_i^{k,j}, \tilde{a}_i^{k,j}) = (s,a) \right] \leq 2(1 + \gamma^2 \mathsf{Var}_P(V)(s,a;w)) \qquad (12)$$

**Lemma 9.** *In the tabular setting, suppose $w$ is a fixed vector such that $\langle w, \phi(s,a) \rangle \in [0, \frac{1}{1-\gamma}]$ for every $(s,a) \in \mathcal{S} \times \mathcal{A}$. Fix $(s,a) \in \mathcal{S} \times \mathcal{A}$. Then there exists a universal constant $C$ such that with probability atleast $1 - \delta$, we have that:*

*Then,*

$$\left| \langle \phi(s,a), \sum_{j=1}^{N} \prod_{l=N}^{j+1} \tilde{H}_{1,B}^{k,l} \tilde{L}^{k,j}(w) \rangle \right| \leq C \sqrt{\eta \log(2/\delta)(1 + \gamma^2 \mathsf{Var}_P(V)(s,a;w))} + C \frac{\eta \log(2/\delta)}{1-\gamma}$$

### G.2   ZIBEL MDP CASE

We will use an appropriate exponential super-martingale to bound the error term in the ZIBEL MDP case just like in the proof of (Jain et al., 2021b, Lemma 27). The following thereom summarizes the result and we refer to Section L.3 for its proof.

**Theorem 4.** *Suppose $x, w \in \mathbb{R}^d$ are fixed. Then, there exists a universal constant $C$ such that with probability at least $1 - \delta$, we have:*

$$\left| \langle x, \sum_{j=1}^{N} \prod_{l=N}^{j+1} \tilde{H}_{1,B}^{k,l} \tilde{L}^{k,j}(w) \rangle \right| \leq C\|x\| (1 + \|w\|_\phi) \sqrt{\eta \log(2/\delta)}. \qquad (13)$$

By a direct application of (Vershynin, 2018, Theorem 8.1.6), we derive the following corollary. We remind the reader that $C_\Phi$ is the covering number defined in Definition 3.

**Corollary 1.** *Suppose $x, w \in \mathbb{R}^d$ are fixed. Then, there exists a universal constant $C$ such that with probability at least $1 - \delta$, we have:*

$$\left\| \sum_{j=1}^{N} \prod_{l=N}^{j+1} \tilde{H}_{1,B}^{k,l} \tilde{L}^{k,j}(w) \right\|_\phi \leq C(1 + \|w\|_\phi)\sqrt{\eta} \left[ C_\Phi + \sqrt{\log(\frac{2}{\delta})} \right].$$

In order to apply the theorem above, we will need to control $\|w_1^{k,1}\|_\phi$ uniformly for all $k$. The following lemma presents such a bound and we refer to Section L.4 for the proof.

**Lemma 10.** *Suppose $\tilde{w}_1^{k,1}$ are the iterates of Q-Rex with coupled data from a ZIBEL MDP. There exist universal constants $C, C_1, C_2$ such that whenever $NB > C_1 \frac{\kappa}{\eta} \log\left(\frac{K\kappa}{\delta(1-\gamma)}\right)$, $\eta < C_2 \frac{(1-\gamma)^2}{C_\phi^2 + \log(K/\delta)}$ and $\eta B < \frac{1}{4}$, with probability at-least $1 - \delta$, the following hold:*

1. *For every $1 \le k \le K$, $\|\tilde{w}_1^{k,1}\|_\phi \le \frac{4}{1-\gamma}$*

2. *For every $1 \le k \le K$, $\|\tilde{\epsilon}_k\|_\phi \le \sqrt{\frac{25K\kappa}{\delta(1-\gamma)^2}} \exp(-\frac{\eta NB}{\kappa}) + \frac{C\sqrt{\eta}}{(1-\gamma)}\left[C_\Phi + \sqrt{\log(\frac{2K}{\delta})}\right]$*

## H    PROOF OF THEOREM 1

*Proof.* Consider the Q learning iteration:

$$\tilde{w}_1^{k+1,1} = \mathcal{T}(\tilde{w}_1^{k,1}) + \tilde{\epsilon}_k \,.$$

Using Lemma 4, we conclude: $\tilde{w}_1^{k+1,1} - w^* = \mathcal{T}(\tilde{w}_1^{k,1}) - \mathcal{T}(w^*) + \epsilon_k$ and thence:

$$\|\tilde{w}_1^{k+1,1} - w^*\|_\phi \le \gamma\|\tilde{w}_1^{k,1} - w^*\|_\phi + \sup_{l \le K}\|\epsilon_l\|_\phi \,.$$

Unrolling the recursion above, we conclude:

$$\|\tilde{w}_1^{K+1,1} - w^*\|_\phi \le \gamma^K\|w^*\|_\phi + \frac{\sup_{l \le K}\|\epsilon_l\|_\phi}{1 - \gamma}$$

Now, we invoke item 2 of Lemma 10 along with the constraints on $N, B, K, \eta$ and Lemma 2 to conclude the result.

$\square$

## I    PROOF OF THEOREM 2

In this section we analyze the output of Q-Rex in the tabular setting and obtain convergence guarantees. To connect with the standard theory for tabular MDP Q-learning in (Li et al., 2021), let us use the standard $Q$-function notation where we assume for all $k$, $\tilde{Q}_1^{k,1} \in \mathbb{R}^{\mathcal{S} \times \mathcal{A}}$ and we have that $\tilde{Q}_1^{k,1}(s,a) = \langle \tilde{w}_1^{k,1}, \phi(s,a) \rangle$. Since $\phi(s,a)$ are the standard basis vectors, we must have $\tilde{Q}_1^{k,1} = \tilde{w}_1^{k,1}$. In the tabular setting, we see by using Lemmas 5, 6, 7, and 9 that for any $\delta > 0$, there exists a universal constant $C$, whenever $NB \ge C\frac{\tau_{\text{mix}}}{\mu_{\text{min}}}\log(\frac{|\mathcal{S}||\mathcal{A}|K}{\delta})$, with probability at-least $1 - \delta$, for every $k \in [K]$ and every $(s,a) \in \mathcal{S} \times \mathcal{A}$:

$$\tilde{Q}_1^{k+1,1} = \mathcal{T}\left[\tilde{Q}_1^{k,1}\right] + \tilde{\epsilon}_k \,, \tag{14}$$

where $\tilde{\epsilon}_k \in \mathbb{R}^{\mathcal{S} \times \mathcal{A}}$ is such that for all $(s,a)$,

$$|\tilde{\epsilon}_k(s,a)| \le C\sqrt{\eta\log\left(\frac{K|\mathcal{S}||\mathcal{A}|}{\delta}\right)\left(1 + \gamma^2\text{Var}_P(V)(s,a;\tilde{Q}_1^{k,1})\right)} + C\frac{\eta\log(\frac{K|\mathcal{S}||\mathcal{A}|}{\delta})}{1 - \gamma}$$

$$+ (1 - \eta)^{\frac{\mu_{\text{min}}NB}{2}}\left\|\tilde{Q}_1^{k,1} - \mathcal{T}\left[\tilde{Q}_1^{k,1}\right]\right\|_\infty \,. \tag{15}$$

Now that we have set-up the notation, we will roughly follow the analysis methods used in (Li et al., 2021). Since we start our algorithm with $\tilde{Q}_1^{1,1} = 0$, and $r_t \in [0,1]$ almost surely, we can easily show that $\tilde{Q}_1^{k,1}(s,a) \in [0, \frac{1}{1-\gamma}]$ for every $k, s, a$. Therefore, we upper bound $\left\|\tilde{Q}_1^{k,1} - \mathcal{T}\left[\tilde{Q}_1^{k,1}\right]\right\|_\infty \le \frac{1}{1-\gamma}$

in Equation (15) to conclude that with probability at-least $1 - \delta$, for every $k \in [K]$ and every $(s, a) \in \mathcal{S} \times \mathcal{A}$:

$$|\tilde{\epsilon}_k(s, a)| \leq C\sqrt{\eta \log(\tfrac{K|\mathcal{S}||\mathcal{A}|}{\delta})\gamma^2 \left[\mathsf{Var}_P(V_k)(s, a)\right]} + \alpha_\eta \tag{16}$$

Where $\alpha_\eta := C\dfrac{\eta \log\left(\tfrac{K|\mathcal{S}||\mathcal{A}|}{\delta}\right)}{1 - \gamma} + C\sqrt{\eta \log\left(\tfrac{K|\mathcal{S}||\mathcal{A}|}{\delta}\right)} + \dfrac{\exp\left(-\tfrac{\eta \mu_{\min} NB}{2}\right)}{1 - \gamma}$

Now we define $\Delta_k = \tilde{Q}_1^{k,1} - Q^*$ and $\pi_k : \mathcal{S} \to \mathcal{A}$ to be the deterministic policy given by $Q_1^{k,1}$ i.e, $\pi_k(s) := \arg\sup_{a \in \mathcal{A}} \tilde{Q}_1^{k,1}(s, a)$ and $\pi^*$ to be optimal policy given by $\pi^*(s) := \arg\sup_{a \in \mathcal{A}} Q^*(s, a)$. We use the convention that we pick a single maximizing action using some rule whenever there are multiple. Similarly, we let $P^{\pi_k}$, to be the Markov transition kernel over $\mathcal{S} \times \mathcal{A}$ given by $P^{\pi_k}((s, a), (s', a')) = P(s'|s, a)\mathbb{1}(a' = \pi_k(s'))$. Similarly, we define $P^{\pi^*}$ with respect to the policy $\pi^*$. It is easy to show that:

$$\mathcal{T}(\tilde{Q}_1^{k,1}) = R + \gamma P^{\pi_k} \tilde{Q}_1^{k,1}.$$

Similarly,

$$Q^* = \mathcal{T}(Q^*) = R + \gamma P^{\pi^*} Q^*.$$

Furthermore given any $Q \in \mathbb{R}^{\mathcal{S} \times \mathcal{A}}$, letting $\pi_Q$ being the greedy policy with respect to the function $Q$ we have that for any policy $\pi$, it can be easily seen from the definitions that following element-wise inequality follows:

$$P^\pi Q \leq P^{\pi_Q} Q.$$

We now use Equation (14) along with the equations above to conclude:

$$\gamma P^{\pi^*}\Delta_k + \tilde{\epsilon}_k \leq \Delta_{k+1} \leq \gamma P^{\pi_k}\Delta_k + \tilde{\epsilon}_k. \tag{17}$$

Here, the inequality is assumed to be point-wise. By properties of Markov transition kernels, we can write:

$$\sum_{k=1}^{K} \gamma^{K-k} \left(P^{\pi^*}\right)^{K-k} \tilde{\epsilon}_k + \gamma^K \left(P^{\pi^*}\right)^K \Delta_1 \leq \Delta_{K+1}$$

$$\leq \sum_{k=1}^{K} \gamma^{K-k} \left(\prod_{l=K}^{k+1} P^{\pi_l}\right) \tilde{\epsilon}_k + \gamma^K \left(\prod_{l=K}^{1} P^{\pi_l}\right) \Delta_1. \tag{18}$$

Here, we use the convention that $\prod_{l=K}^{K+1} P^{\pi_l} = I$. We bound the lower bound and the upper bound given in Equation (18) separately in order to bound $\|\Delta_{K+1}\|_\infty$.

We first consider the lower bound. Using (Azar et al., 2013, Lemma 7), we have:

$$\left\|(I - \gamma P^{\pi^*})^{-1}\sqrt{\mathsf{Var}_P(V^*)}\right\|_\infty \leq \sqrt{\frac{2}{(1 - \gamma)^3}}. \tag{19}$$

We also note from (Li et al., 2021, Equation 64) and basic calculations that:

$$\left\|\sqrt{\mathsf{Var}_P(V_k)} - \sqrt{\mathsf{Var}_P(V^*)}\right\|_\infty \leq \sqrt{\|\mathsf{Var}_P(V_k) - \mathsf{Var}_P(V^*)\|_\infty}$$

$$\leq \sqrt{\frac{4}{1 - \gamma}\|\Delta_k\|_\infty}. \tag{20}$$

Using Equations (15) (16), we conclude that there exist universal constants $C, C_1$ such that with probability at least $1 - \delta$ (interpreting the inequalities as element-wise):

$$
\begin{aligned}
\Delta_{K+1} &\geq -\frac{\alpha_\eta + \gamma^K}{1 - \gamma} - C\sqrt{\eta\gamma^2 \log\left(\frac{K|\mathcal{S}||\mathcal{A}|}{\delta}\right)} \sum_{k=1}^K \gamma^{K-k} \left(P^{\pi^*}\right)^{K-k} \sqrt{\mathsf{Var}_P(V_k)} \\
&\geq -\frac{\alpha_\eta + \gamma^K}{1 - \gamma} - C\sqrt{\eta\gamma^2 \log\left(\frac{K|\mathcal{S}||\mathcal{A}|}{\delta}\right)} \sum_{k=1}^K \gamma^{K-k} \left(P^{\pi^*}\right)^{K-k} \sqrt{\mathsf{Var}_P(V^*)} \\
&\quad - C_1 \sqrt{\frac{1}{1-\gamma}} \sqrt{\eta\gamma^2 \log\left(\frac{K|\mathcal{S}||\mathcal{A}|}{\delta}\right)} \sum_{k=1}^K \gamma^{K-k} \sqrt{\|\Delta_k\|_\infty} \\
&\geq -\frac{\alpha_\eta + \gamma^K}{1 - \gamma} - C\sqrt{\eta\gamma^2 \log\left(\frac{K|\mathcal{S}||\mathcal{A}|}{\delta}\right)} (I - \gamma P^{\pi*})^{-1} \sqrt{\mathsf{Var}_P(V^*)} \\
&\quad - C_1 \sqrt{\frac{1}{1-\gamma}} \sqrt{\eta\gamma^2 \log\left(\frac{K|\mathcal{S}||\mathcal{A}|}{\delta}\right)} \sum_{k=1}^K \gamma^{K-k} \sqrt{\|\Delta_k\|_\infty} \\
&\geq -\frac{\alpha_\eta + \gamma^K}{1 - \gamma} - C\sqrt{\frac{\eta\gamma^2}{(1-\gamma)^3} \log\left(\frac{K|\mathcal{S}||\mathcal{A}|}{\delta}\right)} \\
&\quad - C_1 \sqrt{\frac{1}{1-\gamma}} \sqrt{\eta\gamma^2 \log\left(\frac{K|\mathcal{S}||\mathcal{A}|}{\delta}\right)} \sum_{k=1}^K \gamma^{K-k} \sqrt{\|\Delta_k\|_\infty} .
\end{aligned}
\tag{21}
$$

In the above chain, the first inequality follows from Equations (16) (18), the second inequality from Equation (20) and the fourth inequality from Equation (19). For the upper bound consider the following set of equations interpreting them element-wise which hold for a universal constant $C$ and with probability at least $1 - \delta$.

$$
\begin{aligned}
\Delta_{K+1} &\leq \sum_{k=1}^K \gamma^{K-k} \left(\prod_{l=K}^{k+1} P^{\pi_l}\right) \tilde{\epsilon}_k + \gamma^K \left(\prod_{l=K}^{1} P^{\pi_l}\right) \Delta_1 \\
&\leq \frac{\alpha_\eta + \gamma^K}{1 - \gamma} + C\sqrt{\eta\gamma^2 \log\left(\frac{K|\mathcal{S}||\mathcal{A}|}{\delta}\right)} \sum_{k=1}^K \gamma^{K-k} \left(\prod_{l=K}^{k+1} P^{\pi_l}\right) \sqrt{\mathsf{Var}_P(V_k)} \\
&\leq \frac{\alpha_\eta + \gamma^K}{1 - \gamma} + C\sqrt{\eta\gamma^2 \log\left(\frac{K|\mathcal{S}||\mathcal{A}|}{\delta}\right)} \sum_{k=1}^K \gamma^{K-k} \sqrt{\left(\prod_{l=K}^{k+1} P^{\pi_l}\right) \mathsf{Var}_P(V_k)} \\
&= \frac{\alpha_\eta + \gamma^K}{1 - \gamma} + C\sqrt{\eta\gamma^2 \log\left(\frac{K|\mathcal{S}||\mathcal{A}|}{\delta}\right)} \sum_{k=1}^K \gamma^{\frac{K-k}{2}} \gamma^{\frac{K-k}{2}} \sqrt{\left(\prod_{l=K}^{k+1} P^{\pi_l}\right) \mathsf{Var}_P(V_k)} \\
&\leq \frac{\alpha_\eta + \gamma^K}{1 - \gamma} + C\sqrt{\eta\gamma^2 \log\left(\frac{K|\mathcal{S}||\mathcal{A}|}{\delta}\right)} \sqrt{\sum_{k=1}^K \gamma^{K-k}} \sqrt{\sum_{k=1}^K \gamma^{K-k} \left(\prod_{l=K}^{k+1} P^{\pi_l}\right) \mathsf{Var}_P(V_k)} \\
&\leq \frac{\alpha_\eta + \gamma^K}{1 - \gamma} + C\sqrt{\frac{\eta\gamma^2}{1-\gamma} \log\left(\frac{K|\mathcal{S}||\mathcal{A}|}{\delta}\right)} \sqrt{\sum_{k=1}^K \gamma^{K-k} \left(\prod_{l=K}^{k+1} P^{\pi_l}\right) \mathsf{Var}_P(V_k)} \\
&\leq \frac{\alpha_\eta + \gamma^K}{1 - \gamma} + C\sqrt{\frac{\eta\gamma^2}{1-\gamma} \log\left(\frac{K|\mathcal{S}||\mathcal{A}|}{\delta}\right)} \sqrt{\sum_{k=K/2+1}^K \gamma^{K-k} \left(\prod_{l=K}^{k+1} P^{\pi_l}\right) \mathsf{Var}_P(V_k) + \frac{\gamma^{K/2}}{(1-\gamma)^3}} .
\end{aligned}
\tag{22}
$$

In the above chain, the first inequality follows from Equation (18), the second inequality follows from Equation (16), the third inequality follows from Jensen's inequality and noting that $P^\pi$ is a Markov operator, the fourth inequality via Cauchy-Schwartz and the last inequality by noting that that $\|\mathsf{Var}_P(V_k)\|_\infty \leq 1/(1-\gamma)^2$.

It can now be verified that (Li et al., 2021, Lemma 5) applies in our setting to conclude that:

$$\sum_{k=K/2+1}^{K} \gamma^{K-k} \left(\prod_{l=K}^{k+1} P^{\pi_l}\right) \mathsf{Var}_P(V_k) \leq \frac{4}{\gamma^2(1-\gamma)^2}\left(1 + 2 \max_{K/2+1\leq k\leq K} \|\Delta_k\|_\infty\right).$$

Using the equation above, along with Equations (22) and (21), we conclude there exists a universal constant $C$ such that with probability at least $1 - \delta$, we have:

$$\|\Delta_{K+1}\|_\infty \leq \frac{\alpha_\eta + \gamma^K}{1-\gamma} + C\sqrt{\frac{\eta}{(1-\gamma)^3}\log\left(\frac{K|\mathcal{S}||\mathcal{A}|}{\delta}\right)}\sqrt{1 + \max_{K/2+1\leq k\leq K} \|\Delta_k\|_\infty} + \frac{\gamma^{K/2}}{(1-\gamma)} \quad (23)$$

We note that this works with every $K$ replaced with $l$ for any $l \leq K$, importantly under the same event (with probability at least $1 - \delta$) described above for which Equation (23) holds. Define $L = \frac{K}{2^s}$ for some $s = \lceil \log_2(1 + C \log(\frac{1}{1-\gamma}))\rceil$. Under the conditions of the Theorem, i.e, $K > C_2 \frac{1}{(1-\gamma)}\left(\log(\frac{1}{1-\gamma})\right)^2$, we have $\frac{\gamma^{L/2}}{1-\gamma} < 1$. Therefore we conclude from the discussion above that for every $K \geq l \geq L$, we must have:

$$\|\Delta_{l+1}\|_\infty \leq \frac{\alpha_\eta + \gamma^L}{1-\gamma} + C\sqrt{\frac{\eta}{(1-\gamma)^3}\log\left(\frac{K|\mathcal{S}||\mathcal{A}|}{\delta}\right)}\sqrt{1 + \max_{l/2+1\leq k\leq l} \|\Delta_k\|_\infty}$$

$$\leq \frac{\alpha_\eta + \gamma^L}{1-\gamma} + C\sqrt{\frac{\eta\log\left(\frac{K|\mathcal{S}||\mathcal{A}|}{\delta}\right)}{(1-\gamma)^3}} + C\sqrt{\frac{\eta\log\left(\frac{K|\mathcal{S}||\mathcal{A}|}{\delta}\right)}{(1-\gamma)^3}}\sqrt{\max_{l/2+1\leq k\leq l} \|\Delta_k\|_\infty} \quad (24)$$

To analyze this recursion, we have the following lemma which establishes hyper-contractivity, whose proof we defer to Section M.8.

**Lemma 11.** *Suppose $\alpha, \beta \geq 0$. Consider the function $f : \mathbb{R}^+ \to \mathbb{R}^+$ given by $f(u) = \alpha + \beta\sqrt{u}$. Then, $f$ has the unique fixed point: $u^* := \left(\frac{\beta+\sqrt{\beta^2+4\alpha}}{2}\right)^2$. For $t \in \mathbb{N}$, denoting $f^{(t)}$ to be the $t$ fold composition of $f$ with itself, we have for any $u \in \mathbb{R}^+$:*

$$|f^{(t)}(u) - u^*| \leq \beta^{(2-\frac{1}{2^{t-1}})}|u - u^*|^{\frac{1}{2^t}}.$$

Now consider for $0 \leq a < s$,
$$u_a := \sup_{\frac{K}{2^{s-a}}\leq l\leq K} \|\Delta_{l+1}\|_\infty$$

In lemma 11, define $f$ with $\alpha = \frac{\alpha_\eta+\gamma^L}{1-\gamma} + C\sqrt{\frac{\eta\log\left(\frac{K|\mathcal{S}||\mathcal{A}|}{\delta}\right)}{(1-\gamma)^3}}$ and $\beta = C\sqrt{\frac{\eta\log\left(\frac{K|\mathcal{S}||\mathcal{A}|}{\delta}\right)}{(1-\gamma)^3}}$. The fixed point $u^*$ is such that:

$$u^* \leq C\left[\frac{\eta\log\left(\frac{K|\mathcal{S}||\mathcal{A}|}{\delta}\right)}{(1-\gamma)^3} + \frac{\alpha_\eta + \gamma^L}{1-\gamma} + \sqrt{\frac{\eta\log\left(\frac{K|\mathcal{S}||\mathcal{A}|}{\delta}\right)}{(1-\gamma)^3}}\right] \quad (25)$$

Clearly, by Equation (24), we have: $u_a \leq f(u_{a-1})$ and $\|\Delta_{K+1}\| \leq f(u_{s-1})$. By monotonicity of $f(\cdot)$ and the fact that $u_0 \leq \frac{1}{1-\gamma}$ (Lemma 1):

$$u_{s-1} \leq f^{(s-1)}(u_0) \leq f^{(s-1)}\left(\frac{1}{1-\gamma}\right).$$

Therefore,

$$\|\Delta_{K+1}\|_\infty \leq f(u_{s-1}) \leq f^{(s)}\left(\frac{1}{1-\gamma}\right)$$

Applying Lemma 11, we conclude:

$$\|\Delta_{K+1}\|_\infty \le u^* + \beta^{\left(2 - \frac{1}{2^{s-1}}\right)}\left|\frac{1}{1-\gamma} - u^*\right|^{\frac{1}{2^s}} \tag{26}$$

Note that under the constraints on the parameter $\eta$ and $K$ as stated in the Theorem, we must have $u^* \le \frac{1}{(1-\gamma)^3}$. By our choice of $s$, we must have: $|\frac{1}{(1-\gamma)} - u^*|^{\frac{1}{2^s}} \le C'$. Using this in Equation 26 and the fact that $\beta^{\left(2 - \frac{1}{2^{s-1}}\right)} \le \beta + \beta^2$, we conclude that with probability at-least $1 - \delta$, we must have:

$$\|\Delta_{K+1}\| \le C\left[\frac{\eta \log\left(\frac{K|\mathcal{S}||\mathcal{A}|}{\delta}\right)}{(1-\gamma)^3} + \frac{\alpha_\eta + \gamma^L}{1-\gamma} + \sqrt{\frac{\eta \log\left(\frac{K|\mathcal{S}||\mathcal{A}|}{\delta}\right)}{(1-\gamma)^3}}\right]$$

This proves the first part of the theorem. For the second part, we directly substitute the values provided to verify that we indeed obtain $\epsilon$ error.

## J   PROOF OF THEOREM 3

We will now show uniform convergence type result under Assumption 5. For $(s, a) \in \mathcal{S} \times \mathcal{A}$ and $s' \in \mathsf{supp}(P(\cdot|s, a))$. We define the random variables for all $k$:

$$\hat{P}_k(s'|s, a) = \eta \sum_{j=1}^{N} \sum_{i=1}^{B} (1-\eta)^{\tilde{N}_i^{k,j}(s,a)} \mathbb{1}(\tilde{s}_{i+1}^{k,j} = s', \tilde{s}_i^{k,j} = s, \tilde{a}_i^{k,j} = a)$$

$$\bar{P}_k(s'|s, a) := \eta \sum_{j=1}^{N} \sum_{i=1}^{B} (1-\eta)^{\tilde{N}_i^{k,j}(s,a)} P(s'|s, a) \mathbb{1}(\tilde{s}_i^{k,j} = s, \tilde{a}_i^{k,j} = a).$$

**Lemma 12.** *Suppose Assumption 5 holds. Then, with probability at-least $1 - \delta$, we must have for any fixed $(s, a)$:*

$$\sum_{s' \in \mathsf{supp}(P(\cdot|s,a))} |\hat{P}_k(s'|s, a) - \bar{P}_k(s'|s, a)| \le C\sqrt{\eta \bar{d} \log(4/\delta)} + C\eta \bar{d} \log(4/\delta)$$

We refer to Section L.5 for the proof. We now proceed with the proof of Theorem 3. Recall the noiseless iteration $\bar{w}^k$ defined in the discussion following the statement of Lemma 4. We define $\bar{D}_k := w_0^{k,0} - \bar{w}^k$. Observe that we cannot apply Lemma 9 as in the proof of Theorem 2 where we used $w = w_0^{k,0}$ in order to bound $\|\epsilon_k\|_\infty$. This is because of data re-use which causes $w_0^{k,0}$ to depend on the 'variance' term.

However, note that $\bar{w}^k$ is a deterministic sequence and we can apply Lemma 9 and then use the fact that $\bar{w}^k \approx w_0^{k,0}$ to show a similar concentration inequality. To this end, we prove the following lemma:

**Lemma 13.** *In the tabular setting, we have almost surely:*

$$\left|\left\langle \phi(s, a), \sum_{j=1}^{N} \prod_{l=N}^{j+1} \tilde{H}_{1,B}^{k,l} \tilde{L}^{k,j}(w) - \sum_{j=1}^{N} \prod_{l=N}^{j+1} \tilde{H}_{1,B}^{k,l} \tilde{L}^{k,j}(v) \right\rangle\right|$$

$$\le \gamma \|w - v\|_\phi \sum_{s' \in \mathsf{supp}(P(\cdot|s,a))} |\hat{P}_k(s'|s, a) - \bar{P}_k(s'|s, a)| \tag{27}$$

The proof follows from elementary arguments via. the mean value theorem and triangle inequality. We refer to Section M.7 for the proof. We are now ready to give the proof of Theorem 3.

*Proof of Theorem 3.* We proceed with a similar setup as the proof of Theorem 2. Notice that due to data reuse, the noise $\epsilon_k$ in outer loop $k$ is given by $\epsilon_1(\tilde{w}_1^{k,1})$

$$Q_1^{k+1,1} = \mathcal{T}\left[Q_1^{k,1}\right] + \epsilon_1(\tilde{Q}_1^{k,1}).$$

We also define the noiseless Q iteration such that $\bar{Q}_1^{1,1} = \tilde{Q}_1^{1,1}$ and $\bar{Q}_1^{k+1,1} = \mathcal{T}(\bar{Q}_1^{k,1})$. Let $\bar{\Delta}_k :=$ $\tilde{Q}_1^{k,1} - \bar{Q}_1^{k,1}$.

Now, by Lemma 5, we can write down

$$\epsilon_1(\tilde{Q}_1^{k,1})(s,a) = (1-\eta)^{\tilde{N}^1(s,a)}(\tilde{Q}_1^{k,1} - \tilde{Q}^{k,*}) + \langle \phi(s,a), \sum_{j=1}^{N} \prod_{l=N}^{j+1} \tilde{H}_{1,B}^{1,l} \tilde{L}^{1,j}(\tilde{Q}_1^{k,1}) \rangle$$

$$= (1-\eta)^{\tilde{N}^1(s,a)}(\tilde{Q}_1^{k,1} - \tilde{Q}^{k,*}) + \langle \phi(s,a), \sum_{j=1}^{N} \prod_{l=N}^{j+1} \tilde{H}_{1,B}^{1,l} \left( \tilde{L}^{1,j}(\tilde{Q}_1^{k,1}) - \tilde{L}^{1,j}(\bar{Q}_1^{k,1}) \right) \rangle$$

$$+ \langle \phi(s,a), \sum_{j=1}^{N} \prod_{l=N}^{j+1} \tilde{H}_{1,B}^{1,l} \tilde{L}^{1,j}(\bar{Q}_1^{k,1}) \rangle \tag{28}$$

We bound each of the terms above separately. By union bound, the following statements all hold simultaneously with probability at-least $1 - \delta$. By Theorem 7, whenever $NB > C\frac{\tau_{\text{mix}}}{\mu_{\min}} \log(\frac{|\mathcal{S}||\mathcal{A}|}{\delta})$, we must have $\tilde{N}^{(1)}(s,a) \geq \frac{\mu_{\min}}{2}NB$. A simple observation using recursion shows that $\tilde{Q}_1^{k,1}(s,a), \bar{Q}_1^{k,1}(s,a) \in [0, \frac{1}{1-\gamma}]$. Using Lemmas 12 and 13 we conclude that we must have:

$$\sup_{(s,a)\in\mathcal{S}\times\mathcal{A}} \left| \langle \phi(s,a), \sum_{j=1}^{N} \prod_{l=N}^{j+1} \tilde{H}_{1,B}^{1,l} \left( \tilde{L}^{1,j}(\tilde{Q}_1^{k,1}) - \tilde{L}^{1,j}(\bar{Q}_1^{k,1}) \right) \rangle \right|$$

$$\leq C\|\bar{\Delta}_k\|_\infty \left[ \sqrt{\eta\bar{d}\log\left(\frac{|\mathcal{S}||\mathcal{A}|}{\delta}\right)} + \eta\bar{d}\log\left(\frac{|\mathcal{S}||\mathcal{A}|}{\delta}\right) \right] \tag{29}$$

Now observe that $\bar{Q}_1^{k,1}$ is a deterministic sequence. Therefore, using Lemma 9 uniformly for every $k \leq K$ and $(s,a) \in \mathcal{S} \times \mathcal{A}$:

$$\left| \langle \phi(s,a), \sum_{j=1}^{N} \prod_{l=N}^{j+1} \tilde{H}_{1,B}^{1,l} \tilde{L}^{1,j}(\bar{Q}_1^{k,1}) \rangle \right| \leq C\sqrt{\eta\log\left(\frac{|\mathcal{S}||\mathcal{A}|K}{\delta}\right)(1 + \mathsf{Var}_P(\bar{V}_k)(s,a))} + C\frac{\eta\log\left(\frac{|\mathcal{S}||\mathcal{A}|K}{\delta}\right)}{1-\gamma} \tag{30}$$

Where, $\mathsf{Var}_P(\bar{V}_k) := \mathsf{Var}_P(V)(s,a; \bar{Q}_1^{k,1})$.

We now combine all the above with Equation (28) to show that with probability at least $(1 - \delta)$, we must have uniformly for every $k \leq K$ and $(s,a) \in \mathcal{S} \times \mathcal{A}$:

$$|\tilde{\epsilon}_1(\tilde{Q}_1^{k,1})(s,a)| \leq \frac{1}{1-\gamma} \exp\left(-\frac{\eta\mu_{\min}NB}{2}\right) + C\|\bar{\Delta}_k\|_\infty \left[ \sqrt{\eta\bar{d}\log\left(\frac{|\mathcal{S}||\mathcal{A}|}{\delta}\right)} + \eta\bar{d}\log\left(\frac{|\mathcal{S}||\mathcal{A}|}{\delta}\right) \right]$$

$$+ C\sqrt{\eta\log\left(\frac{|\mathcal{S}||\mathcal{A}|K}{\delta}\right)(1 + \mathsf{Var}_P(\bar{V}_k)(s,a))} + C\frac{\eta\log\left(\frac{|\mathcal{S}||\mathcal{A}|K}{\delta}\right)}{1-\gamma} \tag{31}$$

We will now present a crude bound on $\|\bar{\Delta}_k\|_\infty$ to reduce the analysis of Q-RexDaRe to the analysis of Q-Rex.

**Claim 1.** *Whenever $\eta \leq C_1 \frac{(1-\gamma)^2}{\bar{d}\log\left(\frac{|\mathcal{S}||\mathcal{A}|}{\delta}\right)}$, with probability at-least $1 - \delta$, we must have for every $k \leq K$ uniformly:*

$$\|\bar{\Delta}_k\|_\infty \leq C\frac{\exp\left(-\frac{\eta\mu_{\min}NB}{2}\right)}{(1-\gamma)^2} + C\sqrt{\frac{\eta}{(1-\gamma)^4}\log\left(\frac{|\mathcal{S}||\mathcal{A}|K}{\delta}\right)}$$

*Applying this directly to Equation (31), we conclude that with probability at least $(1 - \delta)$, we must have uniformly for every $k \leq K$ and $(s,a) \in \mathcal{S} \times \mathcal{A}$:*

$$|\tilde{\epsilon}_1(\tilde{Q}_1^{k,1})(s,a)| \leq \frac{C}{(1-\gamma)} \exp\left(-\frac{\eta\mu_{\min}NB}{2}\right) + C\frac{\eta\log\left(\frac{|\mathcal{S}||\mathcal{A}|K}{\delta}\right)}{(1-\gamma)^2}\sqrt{\bar{d}}$$
$$+ C\sqrt{\eta\log\left(\frac{|\mathcal{S}||\mathcal{A}|K}{\delta}\right)(1+\mathsf{Var}_P(\bar{V}_k)(s,a))} + C\frac{\eta\log\left(\frac{|\mathcal{S}||\mathcal{A}|K}{\delta}\right)}{1-\gamma} \tag{32}$$

*Proof of Claim 1.* First note that $\mathcal{T}$ is $\gamma$ contractive under the sup norm. Therefore,

$$\|\bar{\Delta}_{k+1}\|_\infty = \left\|\mathcal{T}(\tilde{w}_1^{k,1}) - \mathcal{T}(\bar{w}_1^{k,1}) + \tilde{\epsilon}_k\right\|_\infty$$
$$\leq \left\|\mathcal{T}(\tilde{w}_1^{k,1}) - \mathcal{T}(\bar{w}_1^{k,1})\right\|_\infty + \|\tilde{\epsilon}_k\|_\infty$$
$$\leq \gamma\|\bar{\Delta}_k\|_\infty + \|\tilde{\epsilon}_k\|_\infty \tag{33}$$

In Equation (31), note that $\mathsf{Var}_P(\bar{V}_k)(s,a) \leq \frac{1}{(1-\gamma)^2}$. Therefore, under the conditions of this Claim, we can take $C_1$ small enough so that uniformly for every $(s,a)$ and $k \leq K$ with probability at-least $(1-\delta)$, Equation (31) becomes:

$$\|\tilde{\epsilon}_k(\tilde{Q}_1^{k,1})\|_\infty \leq \frac{\exp\left(-\frac{\eta\mu_{\min}NB}{2}\right)}{1-\gamma} + \|\bar{\Delta}_k\|_\infty\left(\frac{1-\gamma}{2}\right) + C\sqrt{\frac{\eta}{(1-\gamma)^2}\log\left(\frac{|\mathcal{S}||\mathcal{A}|K}{\delta}\right)}.$$

Combining the display above and the fact that $\tilde{\epsilon}_k = \tilde{\epsilon}_1(\tilde{Q}_1^{k,1})$ in Equation (33), we conclude that with probability at-least $1-\delta$, for every $k \leq K$ uniformly:

$$\|\bar{\Delta}_{k+1}\|_\infty \leq \frac{1+\gamma}{2}\|\bar{\Delta}_k\|_\infty + \frac{\exp\left(-\frac{\eta\mu_{\min}NB}{2}\right)}{1-\gamma} + C\sqrt{\frac{\eta}{(1-\gamma)^2}\log\left(\frac{|\mathcal{S}||\mathcal{A}|K}{\delta}\right)} \tag{34}$$

Unrolling the recursion above and using the fact that $\|\bar{\Delta}_1\|_\infty = 0$, we conclude the statement of the claim.

$\square$

We also note that the deterministic iterations $\bar{Q}_1^{k,1}$ converges exponentially in sup norm to $Q^*$ due to $\gamma$ contractivity of $\mathcal{T}$. That is:

$$\|\bar{Q}_1^{k,1} - Q^*\|_\infty \leq \frac{\gamma^k}{(1-\gamma)} \tag{35}$$

We are now ready to connect up with the proof of Theorem 2 with minor modifications. We follow the same analysis as the proof of Theorem 2 but with $\mathsf{Var}_P(\tilde{V}_k)$ replaced with $\mathsf{Var}_P(\bar{V}_k)$. Similar to the proof of Theorem 2, we can control $\mathsf{Var}_P(\bar{V}_k)$ with respect to $\mathsf{Var}_P(V^*)$ and $\|\bar{Q}_1^{k,1} - Q^*\|_\infty$ along with Equation (35). We also replace $\alpha_\eta$ with

$$\alpha_\eta^{\mathsf{dr}} := \frac{C}{(1-\gamma)}\exp\left(-\frac{\eta\mu_{\min}NB}{2}\right) + C\frac{\eta\log\left(\frac{|\mathcal{S}||\mathcal{A}|K}{\delta}\right)}{(1-\gamma)^2}\sqrt{\bar{d}} + C\frac{\eta\log\left(\frac{|\mathcal{S}||\mathcal{A}|K}{\delta}\right)}{1-\gamma}$$

Therefore, we conclude a version of Equation (23):

$$\|\Delta_{K+1}\|_\infty \leq \frac{\alpha_\eta^{\mathsf{dr}} + \gamma^K}{1-\gamma} + C\sqrt{\frac{\eta}{(1-\gamma)^3}\log\left(\frac{K|\mathcal{S}||\mathcal{A}|}{\delta}\right)}\sqrt{1 + \frac{\gamma^{K/2}}{(1-\gamma)}} \tag{36}$$

Where $\Delta_k := Q_1^{k,1} - Q^*$. Using the bounds on the parameters given in the statement of the Theorem, we conclude the result.

$\square$

## K  MINIMAX LOWERBOUNDS FOR TABULAR MDPS

Suppose $\Theta$ denotes the class of all tuples $(\mathcal{M}, \pi)$ where $\mathcal{M}$ is a tabular MDP and $\pi$ is an exploratory policy such that under the policy $\pi$, the MDP achieves a stationary distribution $\mu$ with minimum probability at-least $\mu_{\min}$ (see section 2.1). The rewards are almost surely in the set [0,1]. We receive the stationary sequence $(s_t, a_t, r_t)_{t=1}^{T}$ from MDP $\mathcal{M}$ under policy $\pi$, which we denote as $(s_t, a_t, r_t)_{t=1}^{T} \sim (\mathcal{M}, \pi)$. Let $\mathcal{F}$ be the class of all estimators $f$ which estimate $Q^*$ for the MDP $\mathcal{M}$ with $f((s_t, a_t, r_t)_{t=1}^{T})$. We write the minimax risk as:

$$\mathcal{L}(\Theta, T) := \inf_{f \in \mathcal{F}} \sup_{(\mathcal{M}, \pi) \in \Theta} \mathbb{E}_{(s_t, a_t, r_t)_{t=1}^{T} \sim (\mathcal{M}, \pi)} \| f((s_t, a_t, r_t)_{t=1}^{T}) - Q^* \|_\infty$$

**Theorem 5.** *There exists a constant $C$ such that, for every $\mu_{min} \in (0, 1/4)$, $\gamma \in (0, 1/2)$ and $T \geq C \frac{1}{\mu_{\min}(1-\gamma)}$, we must have:*

$$\mathcal{L}(\theta, T) \geq C \sqrt{\frac{1}{T(1-\gamma)^3 \mu_{\min}}} .$$

*Therefore, we need $T \geq \frac{1}{\epsilon^2 \mu_{\min}(1-\gamma)^3}$ in order to achieve $\epsilon$ error for the Q values.*

*Proof.* We will use Le-Cam's two point method to prove the result. Consider a class of MDPs denoted by MDP$(q, p)$ $(p, q \in [0, 1])$ with state space $\mathcal{S} = \{0, 1\}$, only one possible action, reward function $R : \mathcal{S} \to \mathbb{R}$, $R(s) = s$, and discount factor $\gamma \in [0, 1)$. The transition probability for the MDP is given by $P(0|0) = 1 - q, P(1|0) = q, P(0|1) = p$ and $P(1|1) = 1 - p$.

Under these conditions, the stationary distribution is given by $\mu(0) = \frac{p}{p+q}$ and $\mu(1) = \frac{q}{p+q}$ and the value function can be written as

$$V(1) = \frac{1}{1 - \gamma(1-p) - \gamma p h(q)},$$

$$V(0) = \frac{h(q)}{1 - \gamma(1-p) - \gamma p h(q)}$$

where $h(q) := \frac{\gamma q}{1 - \gamma(1-q)}$. Since there is only one action in each state, the value function coincides with the Q function.

For showing the lower bound, consider two MDPs with parameters $(p_1, q)$ and $(p_2, q)$ with $q < \min\{p_1, p_2\}$ and $p_1, p_2 < 0.5$. The stationary trajectories of length $T$ under the two MDPs, denoted by the random variables $X_{1:T}^{(i)}$ for $i = 1, 2$, satisfies

$$\mathsf{KL}(X_{1:T}^{(2)} \| X_{1:T}^{(1)}) \leq \frac{2q(p_1 - p_2)^2}{(p_1 + q)(p_2 + q)^2 \ln 2} + T \frac{2q(p_1 - p_2)^2}{(p_2 + q)p_1 \ln 2},$$

using $\mathsf{KL}(\mathsf{Ber}(a) \| \mathsf{Ber}(b)) \leq \frac{2(a-b)^2}{b \ln 2}$ whenever $b \leq 1/2$.

Observe that $\mu_{\min}^{(2)} = \mu^{(2)}(1) = \frac{q}{p_2 + q}$ and $\mu_{\min}^{(1)} = \mu^{(1)}(1) = \frac{q}{p_1 + q}$. Combining this with the KL divergence bound above, we conclude that $\mathsf{KL}(X_{1:T}^{(2)} \| X_{1:T}^{(1)}) < \frac{1}{8}$ if we set Now let $p_1 = \frac{1-\gamma}{2}$, $p_2 - p_1 = c\sqrt{\frac{p_1}{T \mu_{\min}^{(1)}}}$ for small enough constant $c$ and $T \gtrsim \frac{p(p+q)}{q}$. (notice that the equation gives $p_2 < 0.5$ when $T$ satisfies the condition above and $p_1 < 0.25$). By Pinsker's inequality, we must have $\mathsf{TV}(X_{1:T}^{(2)}, X_{1:T}^{(1)}) \leq \frac{1}{2}$.

Elementary calculations show that $|V^{(1)}(1) - V^{(2)}(1)| \gtrsim \frac{|p_1 - p_2|}{(1-\gamma)^2} = \frac{C}{(1-\gamma)^2} \sqrt{\frac{p_1}{T \mu_{\min}^{(1)}}}$ since $q < p_1 = \frac{1-\gamma}{2}$. Therefore:

$$|V^{(1)}(1) - V^{(2)}(1)| \gtrsim \sqrt{\frac{1}{T(1-\gamma)^3 \mu_{\min}^{(1)}}}$$

Consider the semi-metric $\rho(x,y) = \left| \frac{1}{1-\gamma(1-x)-\gamma x h(q)} - \frac{1}{1-\gamma(1-y)-\gamma y h(q)} \right|$ for $x, y \in (0,1)$. Let $\mathbb{P}_i$ be the distribution of $X_{1:T}^{(i)}$, and let the distribution be parameterised by $\theta(\mathbb{P}_i) = V^{(i)}(1)$ for $i = 1, 2$. Let $\hat{\theta}$ be any estimator based on the observations. By (Wainwright, 2019c, Chapter 15),

$$\min_{\hat{\theta}} \max_{P \in \{\mathbb{P}_1, \mathbb{P}_2\}} \mathbb{E}_P \left[ |\hat{\theta} - \theta(P)| \right] \geq C \sqrt{\frac{1}{T(1-\gamma)^3 \mu_{\min}^{(1)}}} (1 - \mathsf{TV}(X_{1:T}^{(2)}, X_{1:T}^{(1)}))$$

$$\geq C_1 \sqrt{\frac{1}{T(1-\gamma)^3 \mu_{\min}^{(1)}}} \tag{37}$$

$\implies$ for less than $\epsilon$ error, $T$ has to be at-least $\frac{1}{\mu_{\min}^{(1)} \epsilon^2 (1-\gamma)^3}$.

$q = \frac{\mu_{\min}(1-\gamma)}{1-2\mu_{\min}}, T \gtrsim \frac{1}{(1-\gamma)\mu_{\min}}$, we can ensure that $\mu_{\min}^{(1)} = 2\mu_{\min}$ and $\mu_{\min}^{(2)} \geq \mu_{\min}$, which ensures that MDP($p_1, q$) and MDP($p_2, q$) are both elements of $\Theta$. Therefore, the two point risk in the LHS of Equation (37) lower bounds $\mathcal{L}(\Theta, T)$, which allows us to conclude the result by substituting $\mu_{\min}^{(1)} = \mu_{\min}$. $\qquad\square$

## L  PROOFS OF CONCENTRATION INEQUALITIES

### L.1  PROOF OF LEMMA 8

First, we leverage the techniques established in (Jain et al., 2021b, Lemma 28) to show Lemma 14. The proof follows by a simple re-writing of the proof of the aforementioned lemma which uses a linear approximation (in $\eta$) to $\tilde{H}_{1,B}^{k,j}$. We omit the proof for the sake of clarity.

**Lemma 14.** *Suppose $\eta B < \frac{1}{3}$. Then, the following PSD inequalities hold almost surely:*

$$I - 2\eta \left( 1 + \frac{\eta B}{1-2\eta B} \right) \sum_{i=1}^{B} \tilde{\phi}_i^{k,j} \left( \tilde{\phi}_i^{k,j} \right)^\top \preceq \left( \tilde{H}_{1,B}^{k,j} \right)^\top \tilde{H}_{1,B}^{k,j} \preceq I - 2\eta \left( 1 - \frac{\eta B}{1-2\eta B} \right) \sum_{i=1}^{B} \tilde{\phi}_i^{k,j} \left( \tilde{\phi}_i^{k,j} \right)^\top \tag{38}$$

*Proof of Lemma 8.* We begin by proving the first part. Using Assumption 1 and the fact that the decoupled trajectory $(\tilde{s}, \tilde{a})_t$ is assumed to be mixed at the start of every buffer, we begin by first noting from Lemma 14 that:

$$0 \preceq \mathbb{E}(\tilde{H}_{1,B}^{k,j})^\top \tilde{H}_{1,B}^{k,j} \preceq I - \eta B \mathbb{E}_{(s,a)\sim\mu} \phi(s,a)(\phi(s,a))^\top \preceq (1 - \tfrac{\eta B}{\kappa}) I . \tag{39}$$

Now, observe that:

$$\| \prod_{j=N}^{1} \tilde{H}_{1,B}^{k,j} g \|^2 = g^\top \prod_{j=1}^{N-1} (\tilde{H}_{1,B}^{k,j})^\top \left[ (\tilde{H}_{1,B}^{k,N})^\top \tilde{H}_{1,B}^{k,N} \right] \prod_{j=N-1}^{1} \tilde{H}_{1,B}^{k,j} g \tag{40}$$

Now note that $\tilde{H}_{1,B}^{k,N}$ is independent of $\tilde{H}_{1,B}^{k,j}$ for $j \leq N-1$. Therefore, taking conditional expectation conditioned on $H_{1,B}^{k,j}$ for $j \leq N-1$ in Equation (40) and using Equation (39), we conclude:

$$\mathbb{E} \| \prod_{j=N}^{1} \tilde{H}_{1,B}^{k,j} g \|^2 \leq (1 - \tfrac{\eta B}{\kappa}) \mathbb{E} \| \prod_{j=N-1}^{1} \tilde{H}_{1,B}^{k,j} g \|^2$$

Applying the equation above inductively, we conclude the result.

We now prove Part 2. We apply Markov's inequality to Part 1 along with Lemma 2 to show that with probability at least $1 - \delta$:

$$\| \prod_{j=N}^{1} \tilde{H}_{1,B}^{k,j} g \|_\phi \leq \| \prod_{j=N}^{1} \tilde{H}_{1,B}^{k,j} g \| \leq \frac{\exp\left( -\frac{\eta NB}{\kappa} \right)}{\sqrt{\delta}} \|g\| \leq \exp(-\tfrac{\eta NB}{\kappa}) \sqrt{\tfrac{\kappa}{\delta}} \|g\|_\phi .$$

$\qquad\square$

### L.2   PROOF OF LEMMA 9

*Proof.* We intend to apply Freedman's inequality (Freedman, 1975) like in (Li et al., 2021, Theorem 4), but in an asynchronous fashion and with Markovian data. Here, reverse experience replay endows our problem with the right filtration structure. Using Equation (9), we can write

$$\langle \phi(s,a), \sum_{j=1}^{N} \prod_{l=N}^{j+1} \tilde{H}_{1,B}^{k,l} \tilde{L}^{k,j}(w) \rangle = \eta \sum_{j=1}^{N} \sum_{i=1}^{B} X_i^{k,j} \tag{41}$$

Where $X_i^{k,j} := (1-\eta)^{\tilde{N}_i^{k,j}(s,a)} \tilde{\epsilon}_i^{k,j} \mathbb{1}\big((\tilde{s}_i^{k,j}, \tilde{a}_i^{k,j}) = (s,a)\big)$. This allows us to define the sequence of sigma algebras $\mathcal{F}_i^{k,j} = \sigma((\tilde{s}_m^{k,l}, \tilde{a}_m^{k,l}) : (l > j \text{ and } m \in [B]) \text{ or } (l = j \text{ and } m \le i))$ - that is, it is the sigma algebra of all states and rewards which appeared before and including $(\tilde{s}_i^{k,j}, \tilde{a}_i^{k,j})$ inside the buffer $j$ and all the states in buffers $l > j$. Notice that $\tilde{N}_i^{k,j}(s,a)$ is measurable with respect to the sigma algebra $\mathcal{F}_i^{k,j}$. Using the fact that the buffers are independent, we conclude that: $\mathbb{E}\left[X_i^{k,j}|\mathcal{F}_i^{k,j}\right] = 0$ and

$$\mathbb{E}\left[|X_i^{k,j}|^2|\mathcal{F}_i^{k,j}\right] \le 2\left[1 + \gamma^2 \mathsf{Var}_P(V)(s,a;w)\right] \mathbb{1}\big((\tilde{s}_i^{k,j}, \tilde{a}_i^{k,j}) = (s,a)\big)(1-\eta)^{2\tilde{N}_i^{k,j}(s,a)}.$$

It is also clear from our assumptions that $|X_i^{j,k}| \le \frac{2}{1-\gamma}$ almost surely. Consider the almost sure inequality for the sum of conditional variances:

$$W^k = \sum_{j=1}^{N} \sum_{i=1}^{B} \mathbb{E}\left[|X_i^{k,j}|^2|\mathcal{F}_i^{k,j}\right]$$

$$\le 2\left[1 + \gamma^2 \mathsf{Var}_P(V)(s,a;w)\right] \sum_{j=1}^{K} \sum_{i=1}^{B} \mathbb{1}\big((\tilde{s}_i^{k,j}, \tilde{a}_i^{k,j}) = (s,a)\big)(1-\eta)^{2\tilde{N}_i^{k,j}(s,a)}$$

$$= 2\left[1 + \gamma^2 \mathsf{Var}_P(V)(s,a;w)\right] \sum_{t=0}^{\tilde{N}^k(s,a)-1} (1-\eta)^{2t} \le 2\left(\frac{1 + \gamma^2 \mathsf{Var}_P(V)(s,a;w)}{\eta}\right) \tag{42}$$

We now apply (Li et al., 2021, Equation (144), Theorem 4) with $R = \frac{1}{1-\gamma}$ and $\sigma^2 = 2\left(\frac{1+\gamma^2 \mathsf{Var}_P(V)(s,a;w)}{\eta}\right)$ to conclude the result. $\qquad\square$

### L.3   PROOF OF THEOREM 4

*Proof.* For the sake of convenience, in this proof we will take $\sigma^2 := 4(1 + \|w\|_\phi^2)$. Suppose $\lambda \in \mathbb{R}$. For $1 \le m \le N$ consider:

$$X_m := \eta\lambda^2\sigma^2 \left\langle x, \prod_{l=N}^{N-m+1} \tilde{H}_{1,B}^{k,l} \left(\prod_{l=N}^{N-m+1} \tilde{H}_{1,B}^{k,l}\right)^\top x \right\rangle + \lambda\left\langle x, \sum_{j=N-m+1}^{N} \left(\prod_{l=N}^{j+1} \tilde{H}_{1,B}^{k,l}\right) \tilde{L}^{k,j}(w) \right\rangle$$

$$X_0 := \|x\|^2 \eta\lambda^2\sigma^2$$

Here we use the convention that $\prod_{l=N}^{j+1} \tilde{H}_{1,B}^{k,l} = I$ whenever $j+1 > N$. We claim that the sequence $\exp(X_m)$ forms a super martingale under an appropriate filtration. In this proof only, consider the sigma algebra $\mathcal{F}_m$ to be the sigma algebra of all the state action reward tuples in buffers $N, \dots, N-m+1$ and let $\mathcal{F}_0$ be the trivial sigma algebra. Notice that $\exp(X_m)$ is $\mathcal{F}_m$ measurable.

**Lemma 15.** *Fix $N \ge m \ge 1$. Suppose $Y \in \mathbb{R}^d$ is a $\mathcal{F}_{m-1}$ measurable random vector. Then,*

$$\mathbb{E}\left[\exp\left(\eta\lambda^2\sigma^2\langle Y, \tilde{H}_{1,B}^{k,N-m+1}\left(\tilde{H}_{1,B}^{k,N-m+1}\right)^\top Y\rangle + \lambda\langle Y, \tilde{L}^{k,N-m+1}(w)\rangle\right)\Big|\mathcal{F}_{m-1}\right] \le \exp\left(\eta\lambda^2\sigma^2\|Y\|^2\right)$$

*In particular, taking $Y = \left(\prod_{l=N}^{N-m+2} \tilde{H}_{1,B}^{k,l}\right)^\top x$, we conclude that $\exp(X_m)$ is a super martingale with respect to the filtration $\mathcal{F}_m$*

*Proof of Lemma 15.* In the proof of this lemma, we will drop the superscripts $k, m$ for the sake of convenience and due to conditioning on $\mathcal{F}_{m-1}$, we will treat $Y$ as a constant. Now we define the natural filtration on the buffer under consideration $(\mathcal{G}_i)_{i=1}^B$ where $\mathcal{G}_i$ is the sigma algebra of the all state-action tuples from $(s_1, a_1), \ldots, (s_i, a_i)$ and rewards $(r_h)_{1 \le h < i}$.

Note that by the definition of $\tilde{L}(w)$, we write: $\tilde{L}(w) = \sum_{i=1}^B \eta \tilde{\epsilon}_i(w) \tilde{H}_{1,i-1} \tilde{\phi}_i$. With this in mind, for $h \in [B]$ define $\tilde{L}_h(w) = \sum_{i=1}^h \eta \tilde{\epsilon}_i(w) \tilde{H}_{1,i-1} \tilde{\phi}_i$. Now, $\tilde{L}(w) = \eta \tilde{\epsilon}_B(w) \tilde{H}_{1,B-1} \tilde{\phi}_B + \tilde{L}_{B-1}(w)$

Now, notice that the random variables $\langle Y, \tilde{L}_{B-1}\rangle, \langle Y, \tilde{H}_{1,B-1}\tilde{\phi}_B\rangle$ and $\langle Y, \tilde{H}_{1,B}^{k,m}\left(\tilde{H}_{1,B}^{k,m}\right)^\top Y\rangle$ are $\mathcal{G}_B$ measureable. Furthermore, we must have: $\mathbb{E}\left[\tilde{\epsilon}_B(w)|\mathcal{G}_B\right] = 0$ and $|\tilde{\epsilon}_B(w)| \le 2(1 + \|w\|_\phi) \le \sqrt{2}\sigma$. Therefore, applying conditional Hoeffding's lemma, we have:

$$\mathbb{E}\left[\exp\left(\eta\lambda^2\sigma^2\langle Y, \tilde{H}_{1,B}\left(\tilde{H}_{1,B}\right)^\top Y\rangle + \lambda\langle Y, \tilde{L}(w)\rangle\right)\Big|\mathcal{G}_B\right]$$

$$= \exp\left(\eta\lambda^2\sigma^2\langle Y, \tilde{H}_{1,B}\left(\tilde{H}_{1,B}\right)^\top Y\rangle + \lambda\langle Y, \tilde{L}_{B-1}(w)\rangle\right)\mathbb{E}\left[\exp\left(\lambda\eta\tilde{\epsilon}_B(w)\langle Y, \tilde{H}_{1,B-1}\tilde{\phi}_B\rangle\right)\Big|\mathcal{G}_B\right]$$

$$\le \exp\left(\eta\lambda^2\sigma^2\langle Y, \tilde{H}_{1,B}\left(\tilde{H}_{1,B}\right)^\top Y\rangle + \lambda\langle Y, \tilde{L}_{B-1}(w)\rangle\right)\exp\left(\lambda^2\eta^2\sigma^2|\langle Y, \tilde{H}_{1,B-1}\tilde{\phi}_B\rangle|^2\right)$$

$$\tag{43}$$

In the third step we have used the conditional version of Hoeffding's lemma. Now, consider $Z := (\tilde{H}_{1,B-1})^\top Y$. Clearly,

$$\langle Y, \tilde{H}_{1,B}\left(\tilde{H}_{1,B}\right)^\top Y\rangle + \eta|\langle Y, \tilde{H}_{1,B-1}\tilde{\phi}_B\rangle|^2 = Z^\top \tilde{H}_{B,B}^\top \tilde{H}_{B,B} Z + \eta Z^\top \tilde{\phi}_B(\tilde{\phi}_B)^\top Z$$

$$= Z^\top \left[I - \eta\tilde{\phi}_B(\tilde{\phi}_B)^\top + \|\tilde{\phi}_B\|^2\eta^2\tilde{\phi}_B(\tilde{\phi}_B)^\top\right] Z$$

$$\le \|Z\|^2$$

Here we have used the fact that $\eta < 1$ and $\|\tilde{\phi}_B\| < 1$. Using the bounds above in Equation (43), we conclude:

$$\mathbb{E}\left[\exp\left(\eta\lambda^2\sigma^2\langle Y, \tilde{H}_{1,B}\left(\tilde{H}_{1,B}\right)^\top Y\rangle + \lambda\langle Y, \tilde{L}_B(w)\rangle\right)\Big|\mathcal{G}_B\right]$$

$$\le \exp\left(\eta\lambda^2\sigma^2\langle Y, \tilde{H}_{1,B-1}\left(\tilde{H}_{1,B-1}\right)^\top Y\rangle + \lambda\langle Y, \tilde{L}_{B-1}(w)\rangle\right) \tag{44}$$

Using Equation (44) recursively, we conclude the first statement of the lemma. The last part of the lemma follows easily from the definition of $X_m$. $\qquad\square$

By Lemma 15, we conclude that $\exp(X_m)$ is a super martingale with respect to the filtration $\mathcal{F}_m$. Therefore, we must have:

$$\mathbb{E}\exp(X_N) \le \exp(X_0) = \exp(\eta\|x\|^2\lambda^2\sigma^2).$$

It is also clear that $X_N \ge \lambda\left\langle x, \sum_{j=N-m+1}^N \left(\prod_{l=N}^{j+1}\tilde{H}_{1,B}^{k,l}\right)\tilde{L}^{k,j}(w)\right\rangle$. Applying Chernoff bound with we conclude that the concentration inequality in Equation (13). We can then directly apply (Vershynin, 2018, Theorem 8.1.6) to Equation (13) in order to obtain uniform concentration bounds.

$\qquad\square$

### L.4 PROOF OF LEMMA 10

*Proof of Lemma 10.* By definition of $\tilde{\epsilon}_k$, we have: $\tilde{w}_1^{k+1,1} = \mathcal{T}(\tilde{w}_1^{k,1}) + \tilde{\epsilon}_k$. Therefore, for any $(s,a) \in \mathcal{S} \times \mathcal{A}$, we must have:

$$\langle \phi(s,a), \tilde{w}_1^{k+1,1} \rangle = R(s,a) + \gamma \mathbb{E}_{s' \sim P(\cdot|s,a)} \sup_{a' \in A} \langle \phi(s',a'), \tilde{w}_1^{k,1} \rangle + \langle \phi(s,a), \tilde{\epsilon}_k \rangle.$$

Using the fact that $R(s,a) \in [0,1]$, we conclude:

$$\|\tilde{w}_1^{k+1,1}\|_\phi \leq 1 + \gamma\|\tilde{w}_1^{k,1}\|_\phi + \|\tilde{\epsilon}_k\|_\phi \tag{45}$$

By independence of outer-loops for the coupled data, we note that $\tilde{w}_1^{k,1}$ is independent of the data in buffer $k$. Therefore we can apply Theorem 4 (and resp. Lemma 8) conditionally with $w = \tilde{w}_1^{k,1}$ (and resp. $g = \tilde{w}_1^{k,1} - w^*$), the bias variance decomposition given in Lemma 5 and the bound on $\|w^*\|_\phi$ in Lemma 4 to conclude that with probability at-least $1 - \delta$, for every $k \leq K$:

$$\|\tilde{\epsilon}_k\|_\phi \leq \sqrt{\frac{K\kappa}{\delta}} \exp(-\frac{\eta N B}{\kappa})(\|\tilde{w}_1^{k,1}\|_\phi + \frac{1}{1-\gamma})$$
$$+ C(1 + \|\tilde{w}_1^{k,1}\|_\phi)\sqrt{\eta}\left[C_\Phi + \sqrt{\log(\frac{2K}{\delta})}\right] \tag{46}$$

We now choose constants $C_1$ and $C_2$ in the statement of the Lemma such that Equation 46 implies:

$$\|\tilde{\epsilon}_k\|_\phi \leq \frac{1-\gamma}{2}\|\tilde{w}_1^{k,1}\|_\phi + 1$$

Using the equation above in Equation (45), we conclude:

$$\|\tilde{w}_1^{k+1,1}\|_\phi \leq 2 + \frac{1+\gamma}{2}\|\tilde{w}_1^{k,1}\|_\phi.$$

Unrolling the recursion above and noting $w_1^{1,1} = 0$, we conclude that with probability at-least $1 - \delta$, we must have $\|\tilde{w}_1^{k,1}\|_\phi \leq \frac{4}{1-\gamma}$ for every $k \leq K$. The bound in item 2 follows by using item 1, Equation (46) and the fact that $\tilde{w}_1^{k,1}$ is independent of the data in buffer $k$ due to our coupling. $\qquad \square$

### L.5 PROOF OF LEMMA 12

*Proof.* For the sake of convenience, we will take $\bar{d} = |\text{supp}(P(\cdot|s,a))|$ and index $\text{supp}(P(\cdot|s,a))$ by $[\bar{d}]$. Consider $Y \in \{-1,1\}^{\bar{d}}$. We consider the class of random variables indexed by elements of $\{-1,1\}^{\bar{d}}$:

$$\Delta(Y;s,a) = \sum_{s'} Y_{s'}\left[\hat{P}_k(s'|s,a) - \bar{P}(s'|s,a)\right]$$

The proof proceeds in a similar way to the proof of Lemma 9 via the Freedman inequality. To bring out the similarities we define similar notation. Consider the sequence of sigma algebras $\mathcal{F}_i^{k,j}$ for $i \in [B]$ and $j \in [K]$ as defined in the proof of Lemma 9. We now define

$$X_i^{k,j}(Y) := (1-\eta)^{\tilde{N}_i^{k,j}(s,a)} \sum_{s'} Y_{s'}\left(\mathbb{1}(\tilde{s}_{i+1}^{k,j} = s', \tilde{s}_i^{k,j} = s, \tilde{a}_i^{k,j} = a) - P(s'|s,a)\mathbb{1}(\tilde{s}_i^{k,j} = s, \tilde{a}_i^{k,j} = a)\right).$$

We note that $\mathbb{E}\left[X_i^{k,j}(Y)|\mathcal{F}_i^{k,j}\right] = 0$ and $|X_i^{k,j}(Y)| \leq 2$ almost surely and a simple calculation reveals that: $\mathbb{E}\left[|X_i^{k,j}(Y)|^2|\mathcal{F}_i^{k,j}\right] \leq (1-\eta)^{2\tilde{N}_i^{k,j}}\mathbb{1}(\tilde{s}_i^{k,j} = s, \tilde{a}_i^{k,j} = a)$

It is also clear that: $\Delta(Y;s,a) = \eta\sum_{j=1}^K \sum_{i=1}^B X_i^{k,j}(Y)$. Apply Freedman's concentration inequality, we conclude for any fixed $Y \in \{-1,1\}^{\bar{d}}$ and $(s,a) \in \mathcal{S} \times \mathcal{A}$,

$$\mathbb{P}(|\Delta(Y;s,a)| > C\sqrt{\eta\log(2/\delta)} + \eta\log(2/\delta)) \leq \delta$$

Applying a union bound over all $Y \in -1, 1^d$, we conclude:

$$\mathbb{P}(\sup_{Y \in \{-1,1\}^{\bar{d}}} |\Delta(Y; s, a)| > C\sqrt{\eta \bar{d} \log(4/\delta)} + \eta \bar{d} \log(4/\delta)) \leq \delta.$$

We complete the proof by noting that

$$\sup_{Y \in \{-1,1\}^{\bar{d}}} |\Delta(Y; s, a)| = \sum_{s' \in \mathsf{supp}(P(\cdot|s,a))} |\hat{P}_k(s'|s,a) - \bar{P}_k(s'|s,a)|$$

$\square$

## M    TECHNICAL LEMMAS

### M.1    COUPLING LEMMA

We first introduce some useful notation: Let $D^{k,j}$ (resp. $\tilde{D}^{k,j}$) be the tuple of random variables $(s_i^{k,j}, a_i^{k,j}, r_i^{k,j})_{i=1}^{B+1}$ (resp. $(\tilde{s}_i^{k,j}, \tilde{a}_i^{k,j}, \tilde{r}_i^{k,j})_{i=1}^{B+1}$).

**Lemma 16.** *Suppose*

$$u \geq \begin{cases} C\tau_{\mathsf{mix}} \log(\frac{T}{\delta}) \text{ in the tabular setting} \\ C\tau_{\mathsf{mix}} \log(\frac{T}{C_{\mathsf{mix}}\delta}) \text{ in the general setting} \end{cases} \quad (47)$$

*Then, we can define the sequences $(s_t, a_t, r_t)$ and $(\tilde{s}_t, \tilde{a}_t, \tilde{r}_t)$ on a common probability space such that:*

1. *The tuples $D^{k,j}$ and $\tilde{D}^{k,j}$ have the same distribution for every $(k, j)$,*

2. *The sequence $\tilde{D}^{k,j}$ for $k \leq N, j \leq K$ is i.i.d.*

3. *Equation* (4) *holds*

*Proof.* The proof of this lemma for the tabular case is a rewriting of the the proofs of Lemmas 1,2,3 in (Bresler et al., 2020). For the general state space case, we apply appropriate modifications as pioneered in (Goldstein, 1979). $\square$

### M.2    PROOF OF LEMMA 2

*Proof.* The first part follows from the definitions. For the second part, note that: $\|x\| \geq \|x\|_\phi$ follows from Cauchy-Schwarz inequality and Assumption 1. For the reverse inequality we use Assumption 4 to show that:

$$\|x\|_\phi^2 \geq \mathbb{E}_{(s,a)\sim\mu}\langle\phi(s,a), x\rangle^2 \geq \frac{\|x\|^2}{\kappa}$$

$\square$

### M.3    PROOF OF LEMMA 3

*Proof.* Existence is guaranteed by Definition (2) and uniqueness follows from the assumption that $\mathsf{span}(\Phi) = \mathbb{R}^d$. $\square$

### M.4    PROOF OF LEMMA 4

*Proof.* Suppose $w_0, \tilde{w}_0 \in \mathbb{R}^d$ are arbitrary. Let $w_1 = \mathcal{T}(w_0)$ and $\tilde{w}_1 = \mathcal{T}(\tilde{w}_0)$. For any $\phi(s, a)$, we have:

$$|\langle\phi(s,a), w_1 - \tilde{w}_1\rangle| = \gamma\big|\mathbb{E}_{s'\sim P(\cdot|s,a)} \sup_{a'\in\mathcal{A}} \langle\phi(s',a'), w_0\rangle - \mathbb{E}_{s'\sim P(\cdot|s,a)} \sup_{a'\in\mathcal{A}} \langle\phi(s',a'), \tilde{w}_0\rangle\big| \quad (48)$$

By Assumption 2, for every fixed $s'$ there exist $a'_{\max}(s'), \tilde{a}'_{\max}(s')$ such that

$$\sup_{a' \in \mathcal{A}} \langle \phi(s', a'(s')), w_0 \rangle = \langle \phi(s', \tilde{a}'_{\max}(s')), w_0 \rangle \quad \text{and} \quad \sup_{a' \in \mathcal{A}} \langle \phi(s', a'), \tilde{w}_0 \rangle = \langle \phi(s', \tilde{a}'_{\max}(s')), \tilde{w}_0 \rangle.$$

Therefore for any $s'$ it holds that,

$$\langle \phi(s', \tilde{a}'_{\max}), w_0 - \tilde{w}_0 \rangle \leq \sup_{a' \in \mathcal{A}} \langle \phi(s', a'), w_0 \rangle - \sup_{a' \in \mathcal{A}} \langle \phi(s', a'), \tilde{w}_0 \rangle \leq \langle \phi(s', a'_{\max}), w_0 - \tilde{w}_0 \rangle$$

$$\implies \big| \sup_{a' \in \mathcal{A}} \langle \phi(s', a'), w_0 \rangle - \sup_{a' \in \mathcal{A}} \langle \phi(s', a'), \tilde{w}_0 \rangle \big| \leq \|w_0 - \tilde{w}_0\|_\phi.$$

Combining this with Equation (48), we conclude the $\gamma$-contractivity of $\mathcal{T}$. By contraction mapping theorem, $\mathcal{T}$ has a unique fixed point we conclude that it is $w^*$ by considering $Q(s, a) := \langle w^*, \phi(s, a) \rangle$ and showing that it satisfies the bellman optimality condition in Equation (2). For the norm inequality, we note that since $R(s, a) \in [0, 1]$ by assumption and $w^* = \mathcal{T}(w^*)$, we have: $|\langle \phi(s, a), w^* \rangle| \leq 1 + \gamma \|w^*\|_\phi$. Therefore, $\|w^*\| \leq \frac{1}{1-\gamma}$. The second norm equality follows from Lemma 2. $\qquad\square$

## M.5   PROOF OF LEMMA 5

*Proof.* We write the iteration in the outer-loop $k$ as:

$$\tilde{w}_{i+1}^{k,j} = \left[ I - \eta \tilde{\phi}_{-i}^{k,j} [\tilde{\phi}_{-i}^{k,j}]^\top \right] \tilde{w}_i^{k,j} + \eta \tilde{\phi}_{-i}^{k,j} \left[ \tilde{r}_{-i}^{k,j} + \gamma \sup_{a' \in \mathcal{A}} \langle \tilde{w}_1^{k,1}, \phi(\tilde{s}_{-(i-1)}^{k,j}, a') \rangle \right]$$

Using the fact that $\langle \tilde{w}^{k+1,*}, \tilde{\phi}_{-i}^{k,j} \rangle = \tilde{R}_{-i}^{k,j} + \gamma \mathbb{E}_{s' \sim P(\cdot | \tilde{s}_{-i}^{k,j}, \tilde{a}_{-i}^{k,j})} \sup_{a' \in A} \langle \phi(s', a'), \tilde{w}_1^{k,1} \rangle$, and recalling the notation in (5),(6), we write:

$$\tilde{w}_{i+1}^{k,j} - \tilde{w}^{k+1,*} = \left[ I - \eta \tilde{\phi}_{-i}^{k,j} [\tilde{\phi}_{-i}^{k,j}]^\top \right] \left( \tilde{w}_i^{k,j} - w^{k+1,*} \right) + \eta \tilde{\phi}_{-i}^{k,j} \epsilon_{-i}^{k,j}.$$

Therefore,

$$\tilde{w}_1^{k,j+1} - \tilde{w}^{k+1,*} = \tilde{H}_{1,B}^{k,j} (\tilde{w}_1^{k,j} - \tilde{w}^{k+1,*}) + \tilde{L}^{k,j}$$

Unfurling further, and using the fact that $\tilde{w}_1^{k,N+1} = \tilde{w}_1^{k+1,1}$, we conclude the statement of the lemma.

$\qquad\square$

## M.6   PROOF OF LEMMA 7

We let $\tilde{N}^{k,j}(s, a) := \sum_{i=1}^B \mathbb{1}((\tilde{s}_i^{k,j}, \tilde{a}_i^{k,j}) = (s, a))$. We want to show that the quantity $\sum_{j=1}^N \tilde{N}^{k,j}(s, a)$ concentrates around its expectation. Note that $\mathbb{E}\tilde{N}^{k,j}(s, a) = B\mu(s, a)$ and that $\tilde{N}^{k,j}(s, a)$ are i.i.d. for $j \in [N]$. From the proof of (Paulin, 2015, Theorem 3.4) and the fact that $\gamma_{\mathsf{ps}} \geq \frac{1}{2\tau_{\mathsf{mix}}}$, we conclude:

$$\mathbb{E}\exp(\lambda(\sum_{j=1}^N \tilde{N}^{k,j}(s, a) - B\mu(s, a))) = \prod_{j=1}^N \mathbb{E}\exp(\lambda(\tilde{N}^{k,j}(s, a) - B\mu(s, a)))$$

$$\leq \exp\left( \frac{8N(B + 2\tau_{\mathsf{mix}})\mu(s, a)\tau_{\mathsf{mix}}\lambda^2}{1 - 20\lambda\tau_{\mathsf{mix}}} \right) \qquad (49)$$

Following the Chernoff bound in the proof of (Paulin, 2015, Theorem 3.4), we conclude:

$$\mathbb{P}(|\tilde{N}^k(s, a) - NB\mu(s, a)| > t) \leq 2\exp\left( -\frac{t^2}{16\tau_{\mathsf{mix}}(B + 2\tau_{\mathsf{mix}})K\mu(s, a) + 40t\tau_{\mathsf{mix}}} \right) \qquad (50)$$

Taking $t = \frac{1}{2}NB\mu(s,a)$, whenever $B \geq \tau_{\text{mix}}$ (by assumption), we must have:

$$\mathbb{P}\left(\tilde{N}^k(s,a) \leq \frac{1}{2}NB\mu(s,a)\right) \leq 2\exp\left(-C\frac{KB\mu(s,a)}{\tau_{\text{mix}}}\right)$$

for some constant $C$. Further, via a union bound over all state action pairs $(s,a)$, we conclude the statement of the lemma.

## M.7  PROOF OF LEMMA 13

We first prove a simple consequence of multivariate calculus.

**Lemma 17.** *Let $f : \mathbb{R}^n \to \mathbb{R}$ be defined by $f(x) = \sup_i x_i$. Then, for every $x, y \in \mathbb{R}^n$, there exists $\zeta \in \mathbb{R}^d$ such that $\|\zeta\|_1 = 1$, $\zeta_i \geq 0$ and:*

$$f(x) - f(y) = \langle \zeta, x - y \rangle$$

*Proof.* We consider the log-sum-exp function. Given $L \in \mathbb{R}^+$, define $f_L(x) := \frac{1}{L}\log\left(\sum_{i=1}^n \exp(Lx_i)\right)$. An elementary calculation shows that:

$$\frac{\log(n)}{L} + \sup_i x_i \geq f_L(x) \geq \sup_i x_i$$

Therefore, for any fixed $x$,

$$\lim_{L \to \infty} f_L(x) = f(x).$$

Now, $\nabla f_L(x) = (p_1, \ldots, p_n)$ where $p_i = \frac{\exp(Lx_i)}{\sum_{j=1}^n \exp(Lx_j)}$. Clearly, $\langle \nabla f_L(x), e_i \rangle \geq 0$ and $\|\nabla f_L(x)\|_1 = 1$. By the mean value theorem there exists $\beta_L$ such that:

$$f_L(x) - f_L(y) = \langle \nabla f_L(\beta_L), x - y \rangle. \tag{51}$$

Now, the simplex in $\mathbb{R}^n$ (denoted by $\Delta_n$) is compact. Therefore, there exists a sub-sequence $L_k \to \infty$ such that $\lim_{k \to \infty} \nabla f_{L_k}(\beta_{L_k}) = \zeta \in \Delta_n$. Taking limit along the sub-sequence $L_k$ in Equation (51), we conclude the result. $\square$

*Proof of Lemma 13.* In the tabular setting, we have the following expression using Equation (41):

$$\text{L.H.S} = \left| \eta\gamma \sum_{j=1}^K \sum_{i=1}^B (1-\eta)^{\tilde{N}_i^{k,j}(s,a)} \mathbb{1}\left[(\tilde{s}_i^{k,j}, \tilde{a}_i^{k,j}) = (s,a)\right] \left[\tilde{\epsilon}_i^{k,j}(w) - \tilde{\epsilon}_i^{k,j}(v)\right] \right| \tag{52}$$

Where

$$\tilde{\epsilon}_i^{k,j}(w) := \left[\sup_{a' \in \mathcal{A}} \langle w, \phi(\tilde{s}_{i+1}^{k,j}, a')\rangle - \mathbb{E}_{s' \sim P(\cdot|\tilde{s}_i^{k,j}, \tilde{a}_i^{k,j})} \sup_{a' \in A} \langle \phi(s', a'), w\rangle\right]$$

Now, by an application of Lemma 17, we show that for some $\eta(\cdot|s', w, v) \in \Delta(\mathcal{A})$, depending only on $(s', w, v)$, we have:

$$\sup_{a' \in A} \langle w, \phi(s', a'), w\rangle - \sup_{a' \in A} \langle w, \phi(s', a'), v\rangle = \sum_{a' \in \mathcal{A}} \zeta(a'|s', w, v)\langle \phi(s', a'), w - v\rangle$$

Now, using the definition of $\hat{P}_k(\cdot|s,a)$ and $\bar{P}(\cdot|s,a)$ in the discussion preceding Lemma 12, we can simplify Equation (52) to:

$$\text{L.H.S.} = \gamma \left| \sum_{\substack{s' \in \text{supp}(P(\cdot|s,a)) \\ a' \in \mathcal{A}}} \left(\hat{P}_k(s'|s,a) - \bar{P}_k(s'|s,a)\right) \zeta(a'|s', w, v)\langle \phi(s', a'), w - v\rangle \right| \tag{53}$$

We conclude the statement of the lemma by an application of the Holder inequality. $\square$

## M.8    PROOF OF LEMMA 11

*Proof.* We solve for $f(u^*) = u^*$ to obtain the relation: $u^* = \alpha + \beta\sqrt{u^*}$, which after discarding the negative solution for $\sqrt{u^*}$ yields the unique solution: $u^* = \left(\frac{\beta+\sqrt{\beta^2+4\alpha}}{2}\right)^2$. To prove the second part, consider for arbitrary $x, y \in \mathbb{R}^+$, the following hyper-contractivity:

$$
\begin{aligned}
|f(x) - f(y)| &= \beta|\sqrt{x} - \sqrt{y}| \\
&\leq \beta\sqrt{|x - y|}
\end{aligned}
\tag{54}
$$

The second step follows from the fact that $|\sqrt{a} - \sqrt{b}| \leq \sqrt{|a - b|}$. Since $u^* = f(u^*)$, we apply the inequality above:

$$
\begin{aligned}
|f^{(t)}(u) - u^*| &= |f^{(t)}(u) - f^{(t)}(u^*)| \\
&\leq \beta\sqrt{|f^{(t-1)}(u) - f^{(t-1)}(u^*)|}
\end{aligned}
\tag{55}
$$

We then conclude the result by induction. $\qquad\square$

