# OpenReview forum: "Online Target Q-learning with Reverse Experience Replay: Efficiently finding the Optimal Policy for Linear MDPs"
_ICLR.cc/2022/Conference — ICLR 2022 Poster_

### Official Review · Reviewer_Da1Z · 2021-10-20

**Correctness:** 4
**Technical Novelty And Significance:** 3
**Empirical Novelty And Significance:** 1
**Recommendation:** 5
**Confidence:** 4

**Main Review:**

Strengths: This paper studies the sample complexity of asynchronous learning. Especially, the new algorithm Q-REX achieves the sample complexity $\frac{|\mathcal{S}||\mathcal{A}|}{\epsilon^{2}(1-\gamma)^{4}}$ and Q-REXDARE achieves complexity $\frac{\max \left(\bar{d}, \frac{1}{\epsilon^{2}}\right)}{\mu_{\min }(1-\gamma)^{3}}$.

Weaknesses:

1. Although Q-learning is known to be sub-optimal, the variant of Q-learning are optimal in a couple of setting. For instance, [1] in online setting and [2] in finite horizon offline setting. Since Q-REXDARE is also a variant of q-learning, I think the improvement is a little bit limited comparing to [2] (which uses variance-reduction for Q-learning with $d_m\approx \mu_\min$). I do notice the there is a theorem 1 which holds for linear MDPs, but I cannot see whether this is optimal. (By the way, the paper did not include the related literature [1],[2])

[1] Zhang et al., Almost Optimal Model-Free Reinforcement Learning via Reference-Advantage Decomposition, NeurIPS 2020,
[2] Yin et al. Near-optimal offline reinforcement learning via double variance reduction, NeurIPS 2021.

2. Another major weakness is the current algorithm can only guarantee Q-function learning, which cannot guarantee $\epsilon$-optimal policy with the same complexity since it is known $\epsilon$-optimal Q-function learning can only imply $(1-\gamma)^{-1}\epsilon$-optimal policy. Therefore, the current result cannot guarantee $\epsilon$-optimal policy with near-optimal sample complexity.




**Summary Of The Paper:**

This paper connects Q-learning with online target learning and reverse experience replay and also obtain the result for linear MDPs. Those results are the first of its kind and are shown to be near-optimal in their respective regime. In particular, for asynchronous setting the Q-REX has order $\frac{|\mathcal{S}||\mathcal{A}|}{\epsilon^{2}(1-\gamma)^{4}}$ and Q-REXDARE has order $\frac{\max \left(\bar{d}, \frac{1}{\epsilon^{2}}\right)}{\mu_{\min }(1-\gamma)^{3}}$. The latter one nearly matches the standard lower bound.

**Summary Of The Review:**

Overall, due to the weaknesses I mentioned above, I think this paper does not cross the bar due to the high selective of ICLR. Since the theorems are correct from my point of view, I will choose weak reject.

---

> ### Author Response · Authors · 2021-11-12
> **Response to the Reviewer**
>
> We thank the reviewer for the detailed review and raising some important concerns. We clarify some of the points raised below.
>
> 1. **Comparison to On-Policy Algorithms**: We will include a discussion regarding comparison to on-policy control algorithms in the next version. We believe it is not fruitful to compare this work with [1]: The reference [1] deals with on-policy control to bound regret via. a UCB type method whereas our work provides guarantees for the practically used Q learning method with offline data. Irrespective of the algorithm, off-policy RL has a different set of issues like instability due to function approximation which is not typically seen with on-policy RL algorithms like SARSA and therefore these works are not comparable. (Please refer to the response to reviewer EMzv for further details)
> 2. **Regarding comparison to [2]:** We will include a reference to [2] in our paper. However, we note that [2] adds explicit variance reduction similar to [Wainwright 2019] and references therein.  Our work demonstrates that there is implicit variance reduction as a consequence of Online Target Learning (which is closely related to Fitted Q iteration [E]) with RER. In fact, one of the main contributions of our work is to show through mathematical analysis that these techniques break spurious correlations and give a natural variance reduction without explicitly adding it.
> 3. **Regarding Lower Bounds:** We agree that Q value recovery is in general weaker than policy recovery. [Azar et. al ‘13] show lower bounds for both Q value error and policy recovery error. We achieve the minimax rates for the former. We note that this is yardstick which is used to compare Q-learning type model free methods in the literature ([Wainwright, 2019B], [Li et al, 2020B], [Li et al, 2021]). We also note that if there is a sub-optimality gap of less than $\epsilon$, then we obtain the exact optimal policy.
>
> [E]  A Theoretical Analysis of Deep Q-Learning

---

> > ### Comment · Reviewer_Da1Z · 2021-11-28
> > **Further question**
> >
> > Dear authors,
> >
> > I acknowledge the authors for producing detailed feedbacks. However, I still have two questions:
> >
> > --- "our work is to show through mathematical analysis that these techniques break spurious correlations and give a natural variance reduction without explicitly adding it" ---
> >
> > It is slightly different from the previous techniques, but why does such an implicit technique matter? Does it provide extra benefit over explicit methods?
> >
> > --- "We agree that Q value recovery is in general weaker than policy recovery." ---
> >
> > May I ask, under the current algorithm, what is the theoretical guarantee for the output policy? It's Ok to be not tight, but can you show me the bound?
> >
> > Lastly, I couldn't see authors include the related literature during the revision period, which makes me less certain about how the next version will look like.
> >
> > I would appreciate it if the authors can further address my concern :)

---

> > > ### Author Response · Authors · 2021-11-28
> > > **Further Clarifications**
> > >
> > > Thank you for the acknowledgement and further questions. We clarify the concerns belows:
> > >
> > > 1. Online target learning is heavily used in practice ([Mnih et al, 2015]) as opposed to other explicit variance reduction techniques. We believe it is fruitful to study the effects and benefits of such methods in order to build a theory which explains the successes of practical RL. This also serves to simplify the space of algorithms which have been proposed efficiently perform RL and to understand what techniques are additionally needed to make RL better in practice. In particular, our result shows adding explicit variance reduction on top of standard RL procedures used in practice might not help much.
> > >
> > > 2. We want to clarify that we agree with the initial observation of the reviewer that an $\epsilon$ approximation of the Q values (as ensured by Theorems 1,2 and 3) implies a $\frac{\epsilon}{1-\gamma}$ sub-optimality of the corresponding policy (see [F]). We will include a discussion of this aspect in the next version of the manuscript. Our previous response clarified why we chose Q value error as the metric for comparison between various Q-learning type methods.
> > >
> > > We apologize for not updating our manuscript with the related literature but we would like to assure the reviewer that we will include citations and a fair comparison in the next version.
> > >
> > > [F] Satinder Singh and Richard Yee. An upper bound on the loss from approximate optimal-value functions. Machine Learning, 16(3):227–233, 1994.

---

### Official Review · Reviewer_JvSF · 2021-10-25

**Correctness:** 4
**Technical Novelty And Significance:** 3
**Empirical Novelty And Significance:** 3
**Recommendation:** 8
**Confidence:** 3

**Main Review:**

In this paper, the authors study the sample efficiency of the Q-learning algorithm, which is a popular approach with broad applications in RL. Based on the standard Q-learning approach, they propose several algorithms called Q-Rex and Q-RexDaRe. With the mixing-time assumption and access to an exploration policy with good coverage, they prove that their algorithms can converge to the optimal policy with near-optimal sample complexity bounds. Their results highlight the benefits of online target learning and experience replay.

Assumption 3 ensures that the Markov chain induced by the policy $\pi$ can quickly converge to a stationary distribution after $O(\tau_{mix})$ steps, and Assumption 4 indicates that the stationary distribution has a good coverage in the feature space. These assumptions simplify the problem since the agent can collect i.i.d. samples from a stationary distribution with good coverage by executing the policy $\pi$. I wonder whether there are concrete examples in which these assumptions hold, or whether these assumptions can be further weaken with other techniques. Besides, I remember that the lower bound proposed in Azar et al. (2013) is proved in the generative-model setting without such assumptions. In other words,  the lower bound in Azar et al. (2013) may not hold with further assumptions in this work, which means that the sample complexity of Q-RexDaRe may not be minimax optimal as claimed by the authors.

The sample complexity (regret) of Q-learning algorithm in the tabular setting has also been studied by Jin et al. [1]. They obtained the near-optimal sample complexity $O(H^4SA/\epsilon^2)$ in the online-exploration setting without access to the exploration policy $\pi$ or any generative model. More comparison between the tabular results in this work and that in [1] is necessary in the related work section.

minor comments:
1. The parameter $\kappa$ may implicitly depends on the feature dimension in the linear setting. Therefore, the sample complexity bounds in the theorems still have implicit dependence on the dimension $d$.
2. Several techniques in this work have also been used in the algorithmic design and complexity analysis of previous related results. For example, the design of gaps with length $u$ between buffers has been applied in [2] to obtain near-optimal regret in the infinite-horizon average-reward setting, and the benefits of experience replay has also been observed in [3] for policy optimization methods. It would be better if these relevant references are added in the paper.

[1] Is Q-learning Provably Efficient?

[2] Model-free Reinforcement Learning in Infinite-horizon Average-reward Markov Decision Processes

[3] Cautiously Optimistic Policy Optimization and Exploration with Linear Function Approximation

**Summary Of The Paper:**

This paper studies the sample efficiency of the Q-learning algorithm in both tabular and linear setting. They propose efficient algorithms which can provably converge to the optimal policy with near-optimal sample complexity bounds.

**Summary Of The Review:**

Overall, the paper is clear with solid theoretical bounds. Their results show that the Q-learning algorithm with online target learning and experience replay can provably converge to the optimal policy with near-optimal sample complexity. One concern is that the assumptions may simplify the analysis of the problem. Besides, more discussion on the related works needs to be added.

---

> ### Author Response · Authors · 2021-11-12
> **Response to the Reviewer**
>
> We thank the reviewer for the detailed feedback. We clarify some of the points raised below.
>
> 1. **Regarding Assumptions 3 and 4:** These assumptions ensure mixing and coverage from a single trajectory of a Markov process which seem critical for $\ell^{\infty}$ norm recovery as considered in the present work. In fact, Theorem 1 in [A] shows that even linear regression with Markovian data, $0$ noise and $\ell^2$ recovery is information theoretically hard when mixing time and condition number are large. Therefore, we believe these are natural assumptions on the data and leave the exploration of alternative conditions for future research.
> Such assumptions are standard and are extensively used in the literature on asynchronous Q learning (see [Li et al, 2020], [B] and references therein). More stringent versions of these assumptions on mixing time and condition numbers have also been used in RL literature on Q learning (for instance, see Assumptions 3.1, 3.2 and the assumptions in Proposition 3.1  in [B] and references therein).   We will clarify these points in the manuscript.
>
> 2. **A Clarification about Mixing Time:** We want to note that the mixing assumption does not make the data look i.i.d. and our novel technique, RER works despite this. For example, suppose that $\tau_{\mathsf{mix}} = 1000$. To make the data look i.i.d, we will have to drop at-least 99.9% of the data and just learn with 0.1% of the data (i.e, consider roughly one every $\tau_{\mathsf{mix}}$ data points). The technical novelty of our method is that we can learn by using at-least 90% of the data due to the clever RER formulation, irrespective of how large the mixing time is. This gives us a sample complexity of the form $O(\tau_{\mathsf{mix}}+\tfrac{1}{\epsilon^2})$ instead of $O(\tfrac{\tau_{\mathsf{mix}}}{\epsilon^2})$, where the latter can be much larger.
>
> 3. **Regarding Lower Bounds:** While [Azar et. al, 2013]’s lower bound holds for any algorithm (not just generative), we agree that the $|S||A|$ type dependence is geared towards a generative model. In particular, it states that every state-action pair has to be sampled $\frac{1}{\epsilon^2(1-\gamma)^3}$ times to learn the Q value to $\epsilon$ error. The reasonable way to compare suchl lower bounds to the much harder off-policy learning is via this visitation frequency argument where $|S||A|$ is replaced with $\frac{1}{\mu_{\min}}$.
>  Elementary calculations adapting the method of [Azar et al, 2013] yields a proof for the minimax lower bound given in the manuscript which holds even for value function approximation. Please see the attachment for a brief sketch. We will include a detailed version of this result in the next version of the manuscript. https://drive.google.com/file/d/1IBvJ5QIOkRwnvYpM6y1aE8a2RcuNnTGr/view?usp=sharing
>
> 4. **Regarding Comparison to On Policy Algorithms:** We will include a discussion on the comparison to on-policy algorithms. However, we note that [1] studies on-policy and episodic RL. This is a very different problem from off-policy Q learning and the comparison is not pertinent. There are several stability issues which are encountered by off-policy RL algorithms which are not present in on-policy algorithms. For instance, SARSA, which is the on-policy variant of Q learning, is stable even with mis-specified linear approximation (See [C],[D]) whereas vanilla Q learning with off-policy data is unstable even for linear MDPs [Baird, 1995]. (We refer to the response to reviewer EMzv for further details).
>
> **Minor comments:**
> 1. We thank the reviewer for pointing out the implicit dimension dependence in $\kappa$. We agree with the reviewer regarding the dimension dependence and we will correct this point. We might still be able to assume a condition number in the subspace of $\mathbb{R}^d$ spanned by the linear embeddings, with dependence only on the dimension of the sub-space.
> 2. We thank the reviewer for the additional references and we will cite them in the next version.
>
> [A] Least Squares Regression with Markovian Data: Fundamental Limits and Algorithms
> [B] Finite-Sample Analysis of Nonlinear Stochastic Approximation with Applications in Reinforcement Learning
> [C] Finite-Sample Analysis for SARSA with Linear Function Approximation
> [D] Reinforcement Learning with Function Approximation Converges to a Region

---

> > ### Comment · Reviewer_JvSF · 2021-11-28
> > **Additional Comments**
> >
> > Dear authors,
> >
> > Sorry for my late reply. I would like to thank the authors for their feedbacks, which clearly address many of my previous concerns. I still have the following additional questions:
> >
> > - The clarification about mixing time sounds reasonable and convincing. The $O(\tau_{min} + 1/\epsilon^2)$ bound looks nice. However, I still feel confused why it is possible to use at least 90% with the help of  the RER formulation, despite the fact that these data cannot be regarded as i.i.d. sampled. Could you please further explain the basic idea to tackle this problem? If it has been already discussed in the paper, please let me know where it is.
> >
> > - I understand that one of the challenges in this work is that the algorithms studied are off-policy in nature (This is also one of the main differences compared with the previous results). The authors also explained the main difficulty of the "deadly triad" in the off-policy setting in their responses. I wonder why "deadly triad" phenomenon disappears in the setting of this work? Is it because of the more powerful representation assumption (linear MDP), or the benign conditions about the mixing time and the data coverage (Assumption 3 and Assumption 4)?

---

> > > ### Author Response · Authors · 2021-11-28
> > > **Further Clarification**
> > >
> > > Thank you for the acknowledgement and further questions. Here are the clarifications:
> > >
> > > 1. It is known that functions with Markovian data (for instance empirical sums of the form $\frac{1}{n}\sum_{i=1}^{n}X_i$) have an extra mixing time dependence in their typical fluctuations (see (Paulin, 2015)) which is tight in the worst case. Such sums arise in the error produced by the algorithm when considering forward-order processing. On the other hand, we show that a natural martingale structure arises with reverse-order processing. Since martingales behave similar to i.i.d. data with respect to concentration (Example: Freedman's inequality, Azuma-Hoeffding inequality), therefore, within the buffer of size $B$, the error produced by the algorithm behaves as though it were produced by i.i.d. data. Atleast 90% of the data is used since the buffer size $B = 10 u$, where $u$ is the gap between the buffers. The gap is introduced to remove dependence between buffers. These aspects are described in Section 1.1. under "Experience Replay and Reverse Experience Replay".
> > >
> > > 2. The deadly triad's destabilizing effect is present even with linear MDPs in vanilla Q learning (Example: [Baird, 1995]). We believe that both the Assumptions 3, 4 and the linear MDP assumption are essential for the stability of our method. Indeed we show that online target learning (OTL) and RER together have a stabilizing effect because a) the linear MDP assumption ensures that the Bellman operator is well defined as a linear function and OTL directly estimates this bellman operator in each outer-loop b) Assumptions 3 and 4 allow RER to estimate the Bellman operator to vanishingly small error, ensuring convergence to optimality.

---

> > > > ### Comment · Reviewer_JvSF · 2021-11-29
> > > > **After Clarification**
> > > >
> > > > Thanks for the clarification. The authors's responses have clearly addressed all my concerns. Therefore, I raise my score from 6  to 8.

---

### Official Review · Reviewer_UL8L · 2021-10-27

**Correctness:** 4
**Technical Novelty And Significance:** 3
**Empirical Novelty And Significance:** 2
**Recommendation:** 8
**Confidence:** 3

**Main Review:**

Strengths:
1. The paper provides solid novel analysis combining the recently proposed reverse experience replay.
2. I could not find no typos and the writing was good.

Weaknesses:
3. Much of the paper is in the appendix (all proofs, the experiments). On the other hand, the paper holds 2 options inside the algorithm (that doesn't seem to be part of the core message but more of a technical point) and 3 algorithms where only one is given and considered in the theory. This makes things unnecessarily complex in the paper itself, while actually missing some of the more interesting complexities of the proofs, or the empirical results.

4. I'm missing some understanding regarding the reverse experience replay. Usually experience replay buffers are used to reduce correlations in the updates of the Q, but when samples are taken serially backwards - won't the correlations be very strong? The observation regarding super-martingale might important here - perhaps it would be better to state it in the paper itself in a more clear way (since this does seem like part of the core of the paper).

5. As a follow-up - I understand the gaps are there to make the buffers independent (also could be made clearer in the paper). But why do we need the buffers to be independent? Is this just as a technicality to make the analysis easier?

**Summary Of The Paper:**

The paper provides sample complexity bounds for a Q-learning based algorithm with target Q and experience replay.
Contributions: The authors provide an algorithm with proven sample complexity for linear MDPs and the tabular setting bridging the current state of the art. The analysis relies on common heuristics of experience replay buffer and using a target network given these some theoretical grounding.

**Summary Of The Review:**

I think the paper provides interesting theoretical results and analysis. The idea of using a reverse experience replay buffer sounds interesting for practical use, though I'm not entirely sold there as mentioned in the main review. The work is not groundbreaking but is a solid advancement in its niche.

---

> ### Author Response · Authors · 2021-11-12
> **Response to the Reviewer**
>
> We thank the reviewer for the kind and detailed review. We clarify some of the points raised below.
>
> 3. We thank the reviewer for pointing out the readability issues with our work. We will incorporate some of the changes which will make the manuscript more readable along the suggested lines.
>
> 4. The correlations are actually removed because of the reverse order processing. We refer to Section 3.1 in [A] for a simple demonstration regarding why this is the case. To give a more RL-related explanation (which is also included in the manuscript): Dynamic programming algorithms for RL process the MDP backwards in time since one needs the knowledge of what happens in the future to understand the best action at current time. We can think of RER as roughly doing the same thing.
>
> 5. The approximate independence is required for technical reasons - we want the algorithm to be approximately streaming (see further motivations in [A]) so that we do not consume too much memory and also use reverse order processing which breaks the spurious correlations. This requires the Buffer-Gap modification we see in the manuscript to obtain RER.
>
> However, we clarify that the independence of buffers is not an assumption. We give a rigorous definition and proof of approximate independence via Goldstein’s maximal couplings for Markov processes in the appendix. (See Section C and Lemma 16 for a formal proof). In particular these coupling methods show that, irrespective of whether we a) run the algorithm on the entire trajectory given as is or b) run the algorithm with independent buffers, the algorithm’s output is the same with high probability.
>
> [A] Streaming Linear Systems Identification with Reverse Experience Replay

---

> > ### Comment · Reviewer_UL8L · 2021-12-06
> > **Thank you for your response**
> >
> > The authors have addressed my concerns and I retain my score.

---

### Official Review · Reviewer_EMzv · 2021-11-02

**Correctness:** 3
**Technical Novelty And Significance:** 2
**Empirical Novelty And Significance:** Not applicable
**Recommendation:** 6
**Confidence:** 3

**Main Review:**

I start by saying that I agree with the motivation of the paper to study practically successful algorithms. However, I am not very convinced with the significance of the results or techniques.

- The paper does not study Q-learning as it is, but uses a fixed-policy to remove correlations between Q-estimate and actions.

- Algorithm 1 seems like a minor variation of variance-reduction technique. I think that the formulation of (reverse) experience-replay or online target Q-learning should be more explicitly described before Algorithm 1.

- Assumption 3 with Assumption 4 is quite strong and crucial for the analysis. With this assumption, I think the analysis should be similar to previous analysis on Q-learning or SARSA (e.g., Zou et al., 2019).

- In Theorem 3, by the sample complexity do you mean $\Theta(NKB)$?

**Summary Of The Paper:**

This paper studies the convergence of Q-learning with two popular empirical heuristics (a) online target learning and (b) experience replay. The analysis in this paper is established upon the off-policy setting along with fast-mixing and minimum reachability assumptions. Main convergence results are provided for the linear MDP where all relevant quantities (transition, reward, and Q functions) can be represented as linear functions. The authors propose two variations of Q-learning, namely, Q-Rex and Q-RexDaRe. Convergence results are also given for two algorithms with the sample-optimality guarantee for Q-RexDaRe.

**Summary Of The Review:**

Overall, I agree that it is important to study practically successful and well-used algorithms (even if they do not have any particular benefits from a theoretical standpoint). However, considering the theoretical nature of the paper, I think the contribution of this paper is marginal given a long line of work in the convergence analysis of practical RL algorithms. I lean toward rejections.

---

> ### Author Response · Authors · 2021-11-12
> **Response to The Reviewer's Concerns**
>
> We thank the reviewer for the detailed reviews. We clarify some of the points raised below.
>
> 1.  **Regarding Basic Setup:** In this paper, we study off-policy Q learning, where it learns the optimal policy via off policy data. The on-policy variant of Q-learning is also popularly known as SARSA(0) which takes the greedy action at every step. Mathematically, off-policy Q-learning suffers from stability issues and SARSA(0) suffers from lack of exploration and therefore the challenges are different. We refer the reviewer to the fourth point regarding the unique technical challenges in off policy RL.
>
> 2. **Regarding Variance Reduction:** We respectfully disagree that Algorithm 1 is a minor variation of the variance reduction technique. Variance-reduced Q learning adds an explicit variance reduction term (see [Wainwright 2019]) whereas our work demonstrates that there is implicit variance reduction as a consequence of Online Target Learning (which is closely related to Fitted Q iteration [E]) with RER. In fact, one of the main contributions of our work is to show through mathematical analysis that these techniques break spurious correlations and give a natural variance reduction without explicitly adding it.
>
> 3. **Regarding Assumptions 3 and 4**:  These assumptions ensure mixing and coverage from a single trajectory of a Markov process which seem critical for $\ell^{\infty}$ norm recovery as considered in the present work. In fact, Theorem 1 in [A] shows that even linear regression with Markovian data, $0$ noise and $\ell^2$ recovery is information theoretically hard when mixing time and condition number are large. Therefore, we believe these are natural assumptions on the data and leave the exploration of alternative conditions for future research.
> Such assumptions are standard and are extensively used in the literature on asynchronous Q learning (see [Li et al, 2020], [B] and references therein). More stringent versions of these assumptions on mixing time and condition numbers have also been used in RL literature on Q learning (for instance, see Assumptions 3.1, 3.2 and the assumptions in Proposition 3.1  in [B] and references therein).   We will clarify these points in the manuscript.
>
> 4. **Regarding Comparison to SARSA:** The comparison to SARSA (Zou et al 2019) is not pertinent in this case. It is a well known fact in RL that Q learning is unstable because of the "deadly triad" of (a) Function approximation (b) Bootstrapping and (c) Off policy learning ([Baird 1995]). While (a) and (b) hold for SARSA, (c) does not hold. In fact, it has been shown that SARSA does not show divergent behavior with linear function approximation (See [D])  whereas Q learning without the on-policy control diverges even when the Bellman operator is exactly representable as a linear function ([Baird 1995]).
> To further elucidate our point: for vanilla Q learning to not diverge to $\infty$ with misspecified linear function approximation (i.e, when the bellman operation cannot be represented exactly as a linear function), stringent assumptions have to be made on the exploration policy already being close to the optimal and these assumptions cannot be removed (Assumption 3.2 and Section 3.3 in [B]). Such assumptions do not have to be made for SARSA (see [C],[D] and references therein) since it is inherently stable.
>
> 5. **Regarding Sample Complexity:** In Theorem 3, $\Theta(NB)$ is in fact the correct sample complexity since Q-RexDaRe (unlike Q-Rex) re-uses data from the first outer-loop iteration.
>
> We respectfully disagree that our work is not theoretically interesting. As we have stressed multiple times in the manuscript, Q learning type methods are known to diverge to infinity even with exact linear function representation. Its convergence guarantees are sample sub-optimal even in the tabular setting. We fix both of these issues with our algorithms Q-Rex and Q-RexDaRe and we believe this is an important theoretical contribution.
>
> [A] Least Squares Regression with Markovian Data: Fundamental Limits and Algorithms
>
> [B] Finite-Sample Analysis of Nonlinear Stochastic Approximation with Applications in Reinforcement Learning
>
> [C] Finite-Sample Analysis for SARSA with Linear Function Approximation
>
> [D] Reinforcement Learning with Function Approximation Converges to a Region
>
> [E] A Theoretical Analysis of Deep Q-Learning

---

> > ### Comment · Reviewer_EMzv · 2021-11-28
> > **Response to Rebuttal**
> >
> > Thanks to the authors for detailed responses. As they well addressed most of my concerns, I raise my score to 6. However, I am still not fully convinced on point 4 -- even if off-policy Q-learning shows an inherently different behavior to SARSA, the analysis in this paper largely relies on the fast-mixing property to make data almost independent across buffers (which is also the backbone of (Zou et al. 2019), and they have done it without the aid of Gap between buffers). I wish I could see what are technically new or interesting points, and they should be more explicitly stressed out.

---

> > > ### Author Response · Authors · 2021-11-28
> > > **More Clarifications**
> > >
> > > We thank the reviewer for the response to our comments.
> > >
> > > **Regarding comparison to results about SARSA:** To further compare the results with [C], we note that the mixing assumptions in [C]  are much more stringent than our setting. In Assumption 1, they require *every* behavior policy to be uniformly exponentially ergodic. We only require the offline policy to be uniformly exponentially ergodic.
> > >
> > > To compare the quantitative results, we refer to Theorem 1 in [C]. The rates they obtain is of the form $\frac{\tau^2_{\mathsf{mix}}}{T}$ (in the specific paper, $\tau_{\mathsf{mix}}$ is denoted by $\tau_0$) and they only obtain $\tilde{O}(\frac{1}{T})$ rates when $T$ is exponentially large compared to the mixing time. Note that this is no better than considering one every mixing time samples to make the data look approximately i.i.d. -- i.e, if the mixing time is $1000$, then we can only use 0.1% of the data and drop 99.9 % to make it look approximately i.i.d.
> > >
> > > The clever RER formulation of our method circumvents this problem and allows us to use 90% of the data as though they were i.i.d. In particular, for the sup-norm bounds, this allows us rates of the form $O(\tau_{\mathsf{mix} }+ \frac{1}{\epsilon^2})$ instead of $O(\frac{\tau_{\mathsf{mix}}}{\epsilon^2})$. We refer to point 2 in the response to reviewer JvSF and further responses for details as to why we do not incur this problem.
> > >
> > > We note that such rates are possible because the linear MDP assumption in our manuscript ensures that the Bellman operator preserves linearity, ensuring that the problem can be reduced to a well-specified linear regression. Such assumptions are not made in [C]. It is not clear whether better rates are possible for the highly mis-specified linear approximations considered in [C] as shown by lower bounds for linear regression with markovian data in [A, Theorem 2]. However, as explained in the manuscript, the linear MDP assumption is now standard in RL literature since it makes the problem more tractable than with arbitrary linear approximations. Our work indeed shows that we can achieve much better performance for RL algorithms with linear MDP assumptions.

---

### Author Response · Authors · 2021-11-18
**Response to the Reviewers**

Dear reviewers,
                         We want to thank you again for the great reviews. We have responded to most of the concerns which were raised and would welcome some feedback regarding this.

---

### Comment · Area_Chair_QEyk · 2021-11-28
**Any final thoughts?**

Dear reviewers,

If you haven't already done so, please read the authors' response, and engage in the final discussions with them. At the very least, acknowledge their responses, and indicate whether they adequately addressed your concerns or not.

November 29th is the end of the discussion period.

Thank you,
Area Chair

---

### Decision · Program_Chairs · 2022-01-20

**Decision:**

Accept (Poster)

**Comment:**

The paper analyzes a variant of the Q-Learning algorithm with two modifications: Online Target Learning (OTL), and Reverse Experience Replay (REP). OTL is essentially the same as using the target network. REP is a new modification of ER, which instead of randomly selecting samples from the buffer, replays them in the reverse order.

Most reviewers are positive about this paper, so I am going to recommend acceptance. There are, however, several concerns that have been raised by the reviewers. As the authors have not revised the paper during the discussion period, my acceptance recommendation is under the good faith expectation that the authors make a serious effort in improving their work based on the reviews. Some of the concerns are:

- The intuition of why REP breaks the correlation is not clear enough. This has been brought up several times by the reviewers.
- What are the technical differences in the analysis compared to previous work such as Zou et al., 2019?
- The kappa appearing in Assumption 4, and showing up in the error bounds, can be dimension dependent. Please clarify this and its effect on the results.
- Much of the paper is in the appendix. It helps if the authors can include more about the proof technique in the main body of the paper.
- Describe the relation between the error in the value function vs. the performance of its greedy policy.